# A Reinforcement Learning Framework for Time Dependent Causal Effects Evaluation in A/B Testing

## Abstract

A/B testing, or online experiment is a standard business strategy to compare a new product with an old one in pharmaceutical, technological, and traditional industries. The aim of this paper is to introduce a reinforcement learning framework for carrying A/B testing in two-sided marketplace platforms, while characterizing the long-term treatment effects. Our proposed testing procedure allows for sequential monitoring and online updating. It is generally applicable to a variety of treatment designs in different industries. In addition, we systematically investigate the theoretical properties (e.g., size and power) of our testing procedure. Finally, we apply our framework to both synthetic data and a real-world data example obtained from a technological company to illustrate its advantage over the current practice.

## 1 Introduction

A/B testing, or online experiment is a business strategy to compare a new product with an old one in pharmaceutical, technological, and traditional industries (e.g., Google, Amazon, or Facebook). Most works in the literature focus on the setting, in which observations are independent across time (see e.g. Johari et al., 2015; 2017, and the references therein). The treatment at a given time can impact future outcomes. For instance, in a ride-sharing company (e.g., Uber), an order dispatching strategy not only affects its immediate income, but also impacts the spatial distribution of drivers in the future, thus affecting its future income. In medicine, it usually takes time for drugs to distribute to the site of action. The independence assumption is thus violated.

The focus of this paper is to test the difference in long-term treatment effects between two products in online experiments. There are three major challenges as follows. (i) The first one lies in modelling the temporal dependence between treatments and outcomes. (ii) Running each experiment takes a considerable time. The company wishes to terminate the experiment as early as possible in order to save both time and budget. (iii) Treatments are desired to be allocated in a manner to maximize the cumulative outcomes and to detect the alternative more efficiently. The testing procedure shall allow the treatment to be adaptively assigned.

We summarize our contributions as follows. First, we introduce a reinforcement learning (RL, see e.g., Sutton & Barto, 2018, for an overview) framework for A/B testing. In addition to the treatment-outcome pairs, it is assumed that there is a set of time-varying state confounding variables. We model the state-treatment-outcome triplet by using the Markov decision process (MDP, see e.g. Puterman, 1994) to characterize the association between treatments and outcomes across time. Specifically, at each time point, the decision maker selects a treatment based on the observed state. The system responds by giving the decision maker a corresponding outcome and moving into a new state in the next time step. In this way, past treatments will have an indirect influence on future rewards through its effect on future state variables. In addition, the long-term treatment effects can be characterized by the value functions (see Section 3.1 for details) that measure the discounted cumulative gain from a given initial state. Under this framework, it suffices to evaluate the difference between two value functions to compare different treatments. This addresses the challenge mentioned in (i).

Second, we propose a novel sequential testing procedure for detecting the difference between two value functions. To the best of our knowledge, this is the first work on developing valid sequential tests in the RL framework. Our proposed test integrates temporal difference learning (see e.g., Precup et al., 2001; Sutton et al., 2008), the $\alpha$-spending approach (Lan & DeMets, 1983) and bootstrap

(Efron & Tibshirani, 1994) to allow for sequential monitoring and online updating. It is generally applicable to a variety of treatment designs, including the Markov design, the alternating-time-interval design and the adaptive design (see Section 4.4). This addresses the challenges in (ii) and (iii).

Third, we systematically investigate the asymptotic properties of our testing procedure. We show that our test not only maintains the nominal type I error rate, but also has non-negligible powers against local alternatives. To our knowledge, these results have not been established in RL.

Finally, we introduce a potential outcome framework for MDP. We state all necessary conditions that guarantee that the value functions are estimable from the observed data.

## 2 RELATED WORK

There is a huge literature on RL such that various algorithms are proposed for an agent to learn an optimal policy and interact with an environment. Our work is closely related to the literature on off-policy evaluation, whose objective is to estimate the value of a new policy based on data collected by a different policy. Popular methods include Thomas et al. (2015); Jiang & Li (2016); Thomas & Brunskill (2016); Liu et al. (2018); Farajtabar et al. (2018); Kallus & Uehara (2019). Those methods required the treatment assignment probability (propensity score) to be bounded away from 0 and 1. As such, they are inapplicable to the alternating-time-interval design, which is the treatment allocation strategy in our real data application.

Our work is related to the temporal-difference learning method based on function approximation. Convergence guarantees of the value function estimators have been derived by Sutton et al. (2008) under the setting of independent noise and by Bhandari et al. (2018) for Markovian noise. However, uncertainty quantification of the resulting value function estimators have been less studied. Such results are critical for carrying out A/B testing. Luckett et al. (2019) outlined a procedure for estimating the value under a given policy. Shi et al. (2020b) developed a confidence interval for the value. However, these methods do not allow for sequential monitoring or online updating.

In addition to the literature on RL, our work is also related to a line of research on evaluating time-varying causal effects (see e.g. Robins, 1986; Boruvka et al., 2018; Ning et al., 2019; Rambachan & Shephard, 2019; Viviano & Bradic, 2019; Bojinov & Shephard, 2020). However, none of the above cited works used an RL framework to characterize treatment effects. In particular, Bojinov & Shephard (2020) proposed to use importance sampling (IS) based methods to test the null hypothesis of no (average) temporal causal effects in time series experiments. Their causal estimand is different from ours since they focused on $p$ lag treatment effects, whereas we consider the long-term effects characterized by the value function. Moreover, their method requires the propensity score to be bounded away from 0 and 1, and thus it is not valid for our applications.

Furthermore, our work is also related to the literature on sequential analysis (see e.g. Jennison & Turnbull, 1999, and the references therein), in particular, the $\alpha$-spending function approach that allocates the total allowable type I error rate at each interim stage according to an error-spending function. Most test statistics in classical sequential analysis have the canonical joint distribution (see Equation (3.1), Jennison & Turnbull, 1999) and their associated stopping boundary can be recursively updated via numerical integration. However, in our setup, test statistics no longer have the canonical joint distribution when adaptive design is used. This is due to the existence of the carryover effects in time. We discuss this in detail in Appendix C. To resolve this issue, we propose a scalable bootstrap-assisted procedure to determine the stopping boundary (see Section 4.3).

Recently, there is a growing literature on bringing classical sequential analysis to A/B testing. In particular, Johari et al. (2015) proposed an always valid test based on the classical mixture sequential probability ratio tests (mSPRT). Kharitonov et al. (2015) propose modified versions of the O'Brien & Fleming and MaxSPRT sequential tests. Deng et al. (2016) studied A/B testing under Bayesian framework. Abhishek & Mannor (2017) developed a bootstrap mSPRT. These tests cannot detect the carryover effects in time, leading to low statistical power in our setup. See the toy examples in Section 4.1 for detailed illustration.

In addition, we note that there is a line of research on bandit/RL with causal graphs (see e.g., Lee & Bareinboim, 2018; 2019). We remark that the problems considered and the solutions developed in this article are different from these works. Specifically, these works considered applying causal inference methods to deal with unmeasured confounders in bandit/RL settings whereas we apply the RL framework to evaluate time-dependant causal effects.

Finally, we relax several key conditions used in Ertefaie (2014) and Luckett et al. (2019) that presented a potential outcome framework for MDP (see Section 3.1 for details). Specifically, Ertefaie (2014) and Luckett et al. (2019) imposed the Markov conditions on the observed data rather than the potential outcomes, while assuming that the outcome at time $t$ is a deterministic function of the state variables at time $t$, $t+1$ and the treatment at time $t$.

## 3 PROBLEM FORMULATION

### 3.1 A POTENTIAL OUTCOME FRAMEWORK FOR MDP

For simplicity, we assume that there are only two treatments (actions, products), coded as 0 and 1, respectively. For any $t \geq 0$, let $\bar{a}_t = (a_0, a_1, \cdots, a_t)^\top \in \{0,1\}^{t+1}$ denote a treatment history vector up to time $t$. Let $\mathbb{S}$ denote the support of state variables and $S_0$ denote the initial state variable. We assume $\mathbb{S}$ is a compact subset of $\mathbb{R}^d$. For any $(\bar{a}_{t-1}, \bar{a}_t)$, let $S_t^*(\bar{a}_{t-1})$ and $Y_t^*(\bar{a}_t)$ be the counterfactual state and counterfactual outcome, respectively, that would occur at time $t$ had the agent followed the treatment history $\bar{a}_t$. The set of potential outcomes up to time $t$ is given by

$$W_t^*(\bar{a}_t) = \{S_0, Y_0^*(a_0), S_1^*(a_0), \cdots, S_t^*(\bar{a}_{t-1}), Y_t^*(\bar{a}_t)\}.$$

Let $W^* = \cup_{t \geq 0, \bar{a}_t \in \{0,1\}^{t+1}} W_t^*(\bar{a}_t)$ be the set of all potential outcomes.

A deterministic policy $\pi$ is a function that maps the space of state variables to the set of available actions. For any such $\pi$, let $\bar{\pi}_t$ denote the treatment history up to time $t$, assigned according to $\pi$. We use $S_t^*(\bar{\pi}_{t-1})$ and $Y_t^*(\bar{\pi}_t)$ to denote the associated potential state and outcome that would occur at time $t$ had the agent followed $\pi$. The goodness of a policy $\pi$ is measured by its value function,

$$V(\pi; s) = \sum_{t \geq 0} \gamma^t \mathrm{E}\{Y_t^*(\bar{\pi}_t)|S_0 = s\},$$

where $0 < \gamma < 1$ is a discounted factor that reflects the trade-off between immediate and future outcomes. Note that our definition of the value function is slightly different from those in the existing literature (see Sutton & Barto, 2018, for example). Specifically, $V(\pi; s)$ is defined through potential outcomes rather than the observed data. Similarly, we define the Q function by

$$Q(\pi; a, s) = \sum_{t \geq 0} \gamma^t \mathrm{E}\{Y_t^*(\bar{\pi}_t(a))|S_0 = s\},$$

where $\{\bar{\pi}_t(a)\}_{t \geq 0}$ denotes the treatment history where the initial action equals to $a$ and all other actions are assigned according to $\pi$.

In our setup, we focus on two nondynamic policies that assign the same treatment at each time point. We use their value functions (denote by $V(1; \cdot)$ and $V(0; \cdot)$) to measure their long-term treatment effects. Meanwhile, our proposed method is equally applicable to the dynamic policy scenario as well. See Section B.1 for details. To quantitatively compare the two policies, we introduce the Average Treatment Effect (ATE) based on their value functions which relates RL to causal inference.

**Definition.** For a given reference distribution function $\mathbb{G}$, ATE is defined by the integrated difference between two value function, i.e., $\mathrm{ATE} = \int_s \{V(1; s) - V(0; s)\}\mathbb{G}(ds)$.

The focus of this paper is to test the following hypotheses:

$$H_0 : \tau_0 = \mathrm{ATE} \leq 0 \quad \text{v.s} \quad H_1 : \tau_0 = \mathrm{ATE} > 0.$$

When $H_0$ holds, the new product is no better than the old one.

### 3.2 IDENTIFIABILITY OF ATE

One of the most important question in causal inference is the identifiability of causal effects. In this section, we present sufficient conditions that guarantee the identifiability of the value function.

In practice, with the exception of $S_0$, the set $W^*$ cannot be observed, whereas at time $t$, we observe the state-action-outcome triplet $(S_t, A_t, Y_t)$. For any $t \geq 0$, let $\bar{A}_t = (A_0, A_1, \cdots, A_t)^\top$ denote the observed treatment history. We first introduce two conditions that are commonly assumed in multi-stage decision making problems (see e.g. Murphy, 2003; Zhang et al., 2013; Kennedy, 2019).

(CA) Consistency assumption: $S_{t+1} = S_{t+1}^*(\bar{A}_t)$ and $Y_t = Y_t^*(\bar{A}_t)$ for all $t \geq 0$, almost surely.

(SRA) Sequential randomization: $A_t$ is independent of $W^*$ given $S_t$ and $\{S_j, A_j, Y_j\}_{0 \leq j < t}$.

The SRA implies that there are no unmeasured confounders and it automatically holds in online experiments, in which the treatment assignment mechanism is pre-specified. In SRA, we allow $A_t$ to depend on the observed data history $S_t, \{S_j, A_j, Y_j\}_{0 \le j < t}$ and thus, the treatments can be adaptively chosen. We next introduce two conditions that are unique to the RL setting.

(MA) Markov assumption: there exists a Markov transition kernel $\mathcal{P}$ such that for any $t \ge 0$, $\bar{a}_t \in \{0,1\}^{t+1}$ and $\mathcal{S} \subseteq \mathbb{R}^d$, we have $\Pr\{S^*_{t+1}(\bar{a}_t) \in \mathcal{S}|W^*_t(\bar{a}_t)\} = \mathcal{P}(\mathcal{S}; a_t, S^*_t(\bar{a}_{t-1}))$.

(CMIA) Conditional mean independence assumption: there exists a function $r$ such that for any $t \ge 0, \bar{a}_t \in \{0,1\}^{t+1}$, we have $\mathrm{E}\{Y^*_t(\bar{a}_t)|S^*_t(\bar{a}_{t-1}), W^*_{t-1}(\bar{a}_{t-1})\} = r(a_t, S^*_t(\bar{a}_{t-1}))$.

These two conditions are central to the empirical validity of reinforcement learning (RL). Specifically, under these two conditions, there exists an optimal policy $\pi^*$ such that $V(\pi^*; s) \ge V(\pi; s)$ for any $\pi$ and $s$. We observe that Ertefaie (2014) and Luckett et al. (2019) imposed the Markov conditions on the observed data rather than the potential outcomes. When CA and SRA hold, these assumptions are equivalent. When SRA is violated, their Markov assumptions could be violated as the treatment depends on unobserved confounders and the observed data process is no longer Markovian. CMIA requires past treatments to affect $Y^*_t(\bar{a}_t)$ only through its impact on $S^*_t(\bar{a}_{t-1})$. In other words, the state variables shall be chosen to include those that serve as important mediators between past treatments and current outcomes. Under MA, CMIA is automatically satisfied when $Y^*_t(\bar{a}_t)$ is a deterministic function of $(S^*_{t+1}(\bar{a}_t), a_t, S^*_t(\bar{a}_{t-1}))$ that measures the system's status at time $t+1$. The latter condition is commonly imposed in the reinforcement learning literature.

To conclude this section, we derive a version of Bellman equation for the Q function under the potential outcome framework. Specifically, for $a' \in \{0,1\}$, let $Q(a'; \cdot, \cdot)$ denote the Q function where treatment $a'$ is repeatedly assigned after the initial decision.

**Lemma 1** *Under MA, CMIA, CA and SRA, for any $t \ge 0$, $a' \in \{0,1\}$ and any function $\varphi : \mathbb{S} \times \{0,1\} \to \mathbb{R}$, we have $E[\{Q(a'; A_t, S_t) - Y_t - \gamma Q(a'; a', S_{t+1})\}\varphi(S_t, A_t)] = 0$.*

*Sketch of Proof: Under MA, CMIA, CA, SRA, the defined Q-function under the potential outcome framework is the same as that defined on the observed data. Lemma 1 thus follows from the classical Bellman equation (see Equation (4.6) in Sutton & Barto, 2018).*

Lemma 1 implies that the Q-function is estimable from the observed data. Specifically, an estimating equation can be constructed based on Lemma 1 and the Q-function can be learned by solving this estimating equation. Note that $V(a, s) = Q(a; a, s)$ and $\tau_0$ is completely determined by the value function $V$. As a result, $\tau_0$ is estimable from the observed data as well. We remark that the positivity assumption is not needed in Lemma 1. Our procedure can thus handle the case where treatments are deterministically assigned, i.e., the behavior policy $b$ is deterministic. This is due to MA and CMIA that assume the system dynamics are invariant across time. To elaborate this, note that the discounted value function is completely determined by the transition kernel $\mathcal{P}$ and the reward function $r$. We remark that these quantities can be consistently estimated under certain conditions (see C1-C3 in Appendix E), regardless of whether $b$ is deterministic or not.

## 4 TESTING PROCEDURE

We first introduce a toy example to illustrate the limitations of existing A/B testing methods. We next present our method and prove its consistency under a variety of different treatment designs.

### 4.1 TOY EXAMPLES

Existing A/B testing methods can only detect short-term treatment effects, but fail to identify any long-term effects. To elaborate this, we introduce two examples below.

**Example 1.** $S_t = 0.5\varepsilon_t$, $Y_t = S_t + \delta A_t$ for any $t \ge 1$ and $S_0 = 0.5\varepsilon_0$.

**Example 2.** $S_t = 0.5S_{t-1} + \delta A_t + 0.5\varepsilon_t$, $Y_t = S_t$ for any $t \ge 1$ and $S_0 = 0.5\varepsilon_0$.

In both examples, the random errors $\{\varepsilon_t\}_{t \ge 0}$ follow independent standard normal distributions and the parameter $\delta$ describes the degree of treatment effects. Suppose $\delta > 0$. Then $H_1$ holds. In Example 1, the observations are independent and there are no carryover effects at all. In this case, both the existing A/B tests and the proposed test are able to discriminate $H_1$ from $H_0$. In Example 2, however, treatments have delayed effects on the outcomes. Specifically, $Y_t$ does not depend on $A_t$, but is affected by $A_{t-1}$ through $S_t$. Existing tests will fail to detect $H_1$ as the short-term conditional

| Example 1 | | | Example 2 | | |
|---|---|---|---|---|---|
| t-test 0.76 | DML-based test 1 | our test 0.98 | t-test 0.04 | DML-based test 0.06 | our test 0.73 |

Table 1: Powers of t-test, DML-based test and the proposed test under Examples 1 and 2, with $T = 500$, $\delta = 0.1$. $\{A_t\}_t$ follow i.i.d. Bernoulli distribution with success probability 0.5.

average treatment effects $E(Y_t|A_t = 1, S_t) - E(Y_t|A_t = 0, S_t) = 0$ in this example. As an illustration, we conduct a small experiment by assuming the decision is made once at $T = 500$, and report the empirical rejection probability of the classical two-sample t-test that is commonly used in online experiments, a more complicated test based on the double machine learning method (DML, Chernozhukov et al., 2017) that is widely employed for inferring causal effects, and the proposed test. It can be seen the competing methods do not have any power under Example 2.

## 4.2 AN OVERVIEW OF THE PROPOSAL

First, we estimate $\tau_0$ based on a version of temporal difference learning. The idea is to apply basis function approximations to solve an estimating equation derived from Lemma 1. Specifically, let $\mathcal{Q} = \{\Psi^\top(s)\beta_{a',a} : \beta_{a',a} \in \mathbb{R}^q\}$ be a large linear approximation space for $Q(a'; a, s)$, where $\Psi(\cdot)$ is a vector containing $q$ basis functions on $\mathbb{S}$. The dimension $q$ is allowed to depend on the number of samples $T$ to alleviate the effects of model misspecification. Let us suppose $Q \in \mathcal{Q}$ for a moment. By Lemma 1, there exists some $\boldsymbol{\beta}^* = (\beta_{0,0}^{*\top}, \beta_{0,1}^{*\top}, \beta_{1,0}^{*\top}, \beta_{1,1}^{*\top})^\top$ such that

$$E[\{\Psi^\top(S_t)\beta_{a',a}^* - Y_t - \gamma\Psi^\top(S_{t+1})\beta_{a',a'}^*\}\Psi(S_t)\mathbb{I}(A_t = a)] = 0, \quad \forall a, a' \in \{0, 1\},$$

where $\mathbb{I}(\cdot)$ denotes the indicator function. Let $\xi(s, a) = \{\Psi^\top(s)\mathbb{I}(a = 0), \Psi^\top(s)\mathbb{I}(a = 1)\}^\top$. The above equations can be rewritten as $E(\boldsymbol{\Sigma}_t\boldsymbol{\beta}^*) = E\boldsymbol{\eta}_t$, where $\boldsymbol{\Sigma}_t$ is a block diagonal matrix given by

$$\boldsymbol{\Sigma}_t = \left[ \begin{array}{cc} \xi(S_t, A_t)\{\xi(S_t, A_t) - \gamma\xi(S_{t+1}, 0)\}^\top & \\ & \xi(S_t, A_t)\{\xi(S_t, A_t) - \gamma\xi(S_{t+1}, 1)\}^\top \end{array} \right]$$

$$\text{and} \quad \boldsymbol{\eta}_t = \{\xi(S_t, A_t)^\top Y_t, \xi(S_t, A_t)^\top Y_t\}^\top.$$

Let $\widehat{\boldsymbol{\Sigma}}(t) = t^{-1}\sum_{j<t}\boldsymbol{\Sigma}_j$ and $\widehat{\boldsymbol{\eta}}(t) = t^{-1}\sum_{j<t}\boldsymbol{\eta}_j$. It follows that $E\{\widehat{\boldsymbol{\Sigma}}(t)\beta^*\} = E\{\widehat{\boldsymbol{\eta}}(t)\}$. This motivates us to estimate $\boldsymbol{\beta}^*$ by $\widehat{\boldsymbol{\beta}}(t) = \{\widehat{\beta}_{0,0}^\top(t), \widehat{\beta}_{0,1}^\top(t), \widehat{\beta}_{1,0}^\top(t), \widehat{\beta}_{1,1}^\top(t)\}^\top = \widehat{\boldsymbol{\Sigma}}^{-1}(t)\widehat{\boldsymbol{\eta}}(t)$. ATE can thus be estimated by the plug-in estimator $\widehat{\tau}(t) = \int_s \Psi^\top(s)\{\widehat{\beta}_{1,1}(t) - \widehat{\beta}_{0,0}(t)\}\mathbb{G}(ds)$.

Second, we use $\widehat{\tau}(t)$ to construct our test statistic at time $t$. We will show $\sqrt{t}\{\widehat{\tau}(t) - \tau_0\}$ is asymptotically normal. Its variance can be consistently estimated by $\widehat{\sigma}^2(t) = \boldsymbol{U}^\top\widehat{\boldsymbol{\Sigma}}^{-1}(t)\widehat{\boldsymbol{\Omega}}(t)\{\widehat{\boldsymbol{\Sigma}}^{-1}(t)\}^\top\boldsymbol{U}$, as $t$ grows to infinity, where $\boldsymbol{U} = \{-\int_{s\in\mathbb{S}}\Psi(s)^\top\mathbb{G}(ds), 0_q^\top, 0_q^\top, \int_{s\in\mathbb{S}}\Psi(s)^\top\mathbb{G}(ds)\}^\top$, $0_q$ denotes a zero vector of length $q$, and $\widehat{\boldsymbol{\Omega}}(t)$ corresponds to some consistent covariance estimator of $\boldsymbol{\eta}_t$ based on the data observed at time $t$ (see equation 3 for the explicit form). This yields our test statistic $\sqrt{t}\widehat{\tau}(t)/\widehat{\sigma}(t)$, at time $t$.

Third, we integrate the $\alpha$-spending approach with bootstrap to sequentially implement our test (see Section 4.3). The idea is to generate bootstrap samples that mimic the distribution of our test statistics, to specify the stopping boundary at each interim stage. Suppose that the interim analyses are conducted at time points $T_1 < \cdots < T_K = T$. For each $1 \le k < K$, we assume $T_k/T \to c_k$ for some constants $0 < c_1 < \cdots < c_{K-1} < 1$.

## 4.3 SEQUENTIAL MONITORING AND ONLINE UPDATING

Let $\{Z_1, \cdots, Z_K\}$ denote the sequence of our test statistics, where $Z_k = \sqrt{T_k}\widehat{\tau}(T_k)/\widehat{\sigma}(T_k)$. To sequentially monitor our test, we need to specify the stopping boundary $\{b_k\}_{1\le k\le K}$ such that the experiment is terminated and $H_0$ is rejected when $Z_k > b_k$ for some $k$.

First, we use the $\alpha$ spending function approach to guarantee the validity of our test. It requires to specify a monotonically increasing function $\alpha(\cdot)$ that satisfies $\alpha(0) = 0$ and $\alpha(T) = \alpha$. Some popular choices of the $\alpha$ spending function include

$$\alpha_1(t) = 2 - 2\Phi\{\Phi^{-1}(1 - \alpha/2)\sqrt{T/t}\} \quad \text{and} \quad \alpha_2(t) = \alpha(t/T)^\theta \quad \text{for} \quad \theta > 0, \tag{1}$$

where $\Phi(\cdot)$ denotes the normal cumulative distribution function. Adopting the $\alpha$ spending approach, we require $b_k$'s to satisfy

$$\Pr(\cup_{j=1}^k\{Z_j > b_j\}) = \alpha(T_k) + o(1), \quad \forall 1 \le k \le K. \tag{2}$$

As commented in the introduction, the numerical integration method is not applicable to determine the stopping boundary. Our method is built upon the wild bootstrap (Wu et al., 1986). The idea is to generate bootstrap samples that have asymptotically the same joint distribution as the test statistics. However, we note that directly applying the wild bootstrap algorithms is time consuming. See Section C for details. To facilitate the computation, we present a scalable bootstrap algorithm to determine $\{b_k\}_k$. Let $\{e_k\}_k$ be a sequence of i.i.d $N(0, I_{4q})$ random vectors, where $I_J$ stands for a $J \times J$ identity matrix for any $J$. Let $\widehat{\boldsymbol{\Omega}}(T_0)$ be an $4q \times 4q$ zero matrix. At the $k$-th stage, we compute

$$\widehat{Z}_k^* = \frac{\boldsymbol{U}^\top \widehat{\boldsymbol{\Sigma}}^{-1}(T_k)}{\sqrt{T_k}\widehat{\sigma}(T_k)} \sum_{j=1}^k \{T_j \widehat{\boldsymbol{\Omega}}(T_j) - T_{j-1}\widehat{\boldsymbol{\Omega}}(T_{j-1})\}^{1/2} e_j.$$

A key observation is that, conditional on the observed dataset, the covariance of $\widehat{Z}_{k_1}^*$ and $\widehat{Z}_{k_2}^*$ equals that of $Z_{k_1}$ and $Z_{k_2}$. See Theorem 3 for details. In addition, the limiting distributions of $\{Z_k\}_k$ and $\{Z_k^*\}_k$ are multivariate normal. As such, the joint distribution of $\{Z_k\}_k$ can be well approximated by that of $\{Z_k^*\}_k$ conditional on the data. This forms the basis of our bootstrap algorithm. By the requirement on $\{b_k\}_k$ in 2, we obtain $\Pr(\max_{1 \le j < k}(Z_j - b_j) \le 0, Z_k > b_k) = \alpha(T_k) - \alpha(T_{k-1}) + o(1)$. To implement our test, we recursively calculate the threshold $\widehat{b}_k$ as follows,

$$\Pr^* \left\{ \max_{1 \le j < k}(Z_j^* - \widehat{b}_j) \le 0, Z_k^* > \widehat{b}_k \right\} = \alpha(T_k) - \alpha(T_{k-1}),$$

where $\Pr^*$ denotes the conditional probability given on the data, and reject $H_0$ when $Z_k^* > \widehat{b}_k$ for some $k$. In practice, the left-hand-side can be approximated via Monte carlo simulations.

### 4.4 CONSISTENCY UNDER DIFFERENT TREATMENT DESIGNS

We consider three treatment allocation designs that can be handled by our procedure as follows:

**D1**. Markov design: $\Pr(A_t = 1|S_t, \{S_j, A_j, Y_j\}_{0 \le j < t}) = b^{(0)}(S_t)$ for some function $b^{(0)}(\cdot)$ uniformly bounded away from 0 and 1.

**D2**. Alternating-time-interval design: $A_{2j} = 0$, $A_{2j+1} = 1$ for all $j \ge 0$.

**D3**. Adaptive design (e.g., $\epsilon$-greedy): For $T_k \le t < T_{k+1}$ for some $k \ge 0$, $\Pr(A_t = 1|S_t, \{S_j, A_j, Y_j\}_{0 \le j < t}) = b^{(k)}(S_t)$ for some $b^{(k)}(\cdot)$ that depends on $\{S_j, A_j, Y_j\}_{0 \le j < T_k}$.

Here, D2 is a deterministic design and is widely used in industry (see our real data example). D1 and D3 are random designs. D1 is commonly assumed in the literature on off-policy evaluation (see e.g., Jiang & Li, 2016). D3 is widely employed in the contextual bandit setting to balance the trade-off between exploration and exploitation. These three settings cover a variety of scenarios in practice.

**Theorem 1 (Type-I error)** *Suppose $\alpha(\cdot)$ is continuous, C1-C3 (see Appendix E) hold and $q = o(\sqrt{T}/\log T)$. Then $Pr(\bigcup_{j=1}^k \{Z_j > \widehat{b}_j\}) \le \alpha(T_k) + o(1)$, for all $1 \le k \le K$ under $H_0$.*

*Sketch of Proof: We consider the case where $\tau_0 = 0$ only. The general case is proven in Section F.3. As discussed in Section 4.3, the conditional distribution of $\{Z_k^*\}_k$ given the data is equivalent as the distribution of $\{Z_k\}_k$. Since $\{\widehat{b}_k\}_k$ is a continuous function of $\{Z_k^*\}_k$, it follows from the continuous mapping theorem that $\{\widehat{b}_k\}_k$ are consistent. The proof is hence completed.*

Theorem 1 implies that the type-I error rate of the proposed test is well controlled. When ATE= 0, the equality in Theorem 1 holds. The rejection probability achieves the nominal level under $H_0$.

**Theorem 2 (Power)** *Under the conditions of Theorem 1, assume $\tau_0 \gg T^{-1/2}$, then $Pr(Z_1 > \widehat{b}_1) \to 1$. Assume $\tau_0 = T^{-1/2}h$ for some $h > 0$. Then $\lim_{T \to \infty}[Pr(\cup_{j=1}^k \{Z_j > \widehat{b}_j\}) - \alpha(T_k)] > 0$.*

*Sketch of Proof: Under $H_1$, similar to Theorem 1, we have $Pr(\cup_{j=1}^k \{Z_j - \sqrt{T_j}\tau_0/\widehat{\sigma}(T_j) > \widehat{b}_j\} = \alpha(T_k) + o(1)$. The assertion follows by that $Z_j$ is stochastically larger than $Z_j - \sqrt{T_j}\tau_0$ for all $j$.*

The second assertion in Theorem 2 implies that our test has non-negligible powers against local alternatives converging to $H_0$ at the $T^{-1/2}$ rate. When the signal decays to zero faster than this rate, our test is not able to detect $H_1$. When the signal decays at a slower rate, the power of our test approaches 1. Combining Theorems 1 and 2 yields the consistency of our test.

Finally, it is worth mentioning that our test can be online updated as batches of observations arrive at the end of each interim stage. We summarize our procedure in Algorithm 1 (see Appendix A). Its time complexity is dominated by $O(Bq^3 + Tq^2)$.

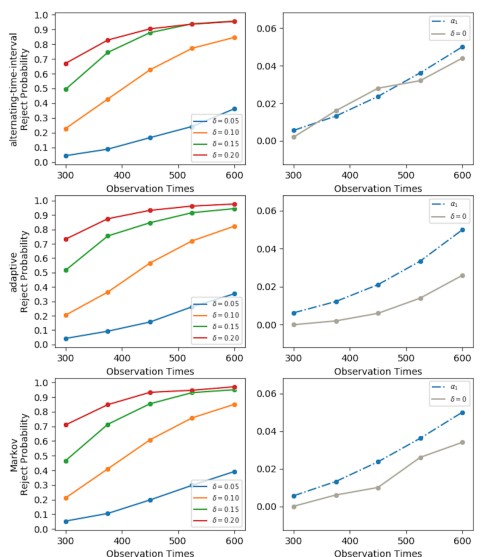 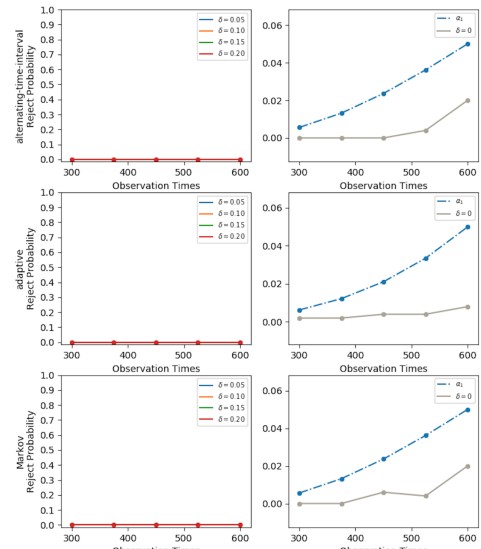

(a) The proposed test under $H_1$ (left) and $H_0$ (right)     (b) Two-sample t-test under $H_1$ (left) and $H_0$ (right)

Figure 1: Empirical rejection probabilities of our test and the two-sample t-test with $\alpha(\cdot) = \alpha_1(\cdot)$. Settings correspond to the alternating-time-interval, adaptive and Markov design, from top plots to bottom plots.

## 5 NUMERICAL STUDIES

### 5.1 SYNTHETIC DATA

Simulated data of states and rewards was generated as follows,

$$S_{1,t} = (2A_{t-1} - 1)S_{1,(t-1)}/2 + S_{2,(t-1)}/4 + \delta A_{t-1} + \varepsilon_{1,t},$$
$$S_{2,t} = (2A_{t-1} - 1)S_{2,(t-1)}/2 + S_{1,(t-1)}/4 + \delta A_{t-1} + \varepsilon_{2,t}, \quad Y_t = 1 + (S_{1,t} + S_{2,t})/2 + \varepsilon_{3,t},$$

where the random errors $\{\varepsilon_{j,t}\}_{j=1,2,0\leq t\leq T}$ are i.i.d $N(0, 0.5^2)$ and $\{\varepsilon_{3,t}\}_{0\leq t\leq T}$ are i.i.d $N(0, 0.3^2)$. The initial states $S_{1,0}$ and $S_{2,0}$ are independent $N(0, 0.5^2)$ as well. Let $S_t = (S_{1,t}, S_{2,t})^\top$ denote the state at time $t$. Under this model, treatments have delayed effects on the outcomes, as in Example 2. The parameter $\delta$ characterizes the degree of such carryover effects. When $\delta = 0$, $\tau_0 = 0$ and $H_0$ holds. When $\delta > 0$, $H_1$ holds. Moreover, $\tau_0$ increases as $\delta$ increases.

We set $K = 5$ and $(T_1, T_2, T_3, T_4, T_5) = (300, 375, 450, 525, 600)$. The discounted factor $\gamma$ is set to 0.6 and $\mathbb{G}$ is chosen as the initial state distribution. We consider three behavior policies, according to the designs D1-D3, respectively. For the behavior policy in D1, we set $b^{(0)}(s) = 0.5$ for any $s \in \mathbb{S}$. For the behavior policy in D3, we use an $\epsilon$-greedy policy and set $b^{(k)}(s) = \epsilon/2 + (1 - \epsilon)\mathbb{I}(\Psi(s)^\top(\widehat{\beta}_{1,1}(T_k) - \widehat{\beta}_{0,0}(T_k)) > 0)$, with $\epsilon = 0.1$, for any $k \geq 1$ and $s \in \mathbb{S}$.

For each design, we further consider five choices of $\delta$, corresponding to $0, 0.05, 0.1, 0.15$ and $0.2$. The significance level $\alpha$ is set to 0.05 in all cases. To implement our test, we choose two $\alpha$-spending functions, corresponding to $\alpha_1(\cdot)$ and $\alpha_2(\cdot)$ given in equation 1. The hyperparameter $\theta$ in $\alpha_2(\cdot)$ is set to 3. The number of bootstrap sample is set to 1000. In addition, we consider the following polynomial basis function, $\Psi(s) = \Psi(s_1, s_2) = (1, s_1, s_1^2, \cdots, s_1^J, s_2, s_2^2, \cdots, s_2^J)^\top$, with $J = 4$. We also tried some other values of $J$ by setting $J$ to 3 and 5. Results are reported in Figure 6 (see Appendix G). It can be seen that the resulting tests is not sensitive to the choice of $J$.

All experiments run on a macbook pro with a dual-core 2.7 GHz processor. Implementing a single test takes one second. Figures 1 (a) and 5 (a) (see Appendix G) depict the empirical rejection probabilities of our test statistics at different interim stages under $H_0$ and $H_1$ with different combinations of $\delta$, $\alpha(\cdot)$ and the designs. These rejection probabilities are aggregated over 500 simulations. We also plot $\alpha_1(\cdot)$ and $\alpha_2(\cdot)$ under $H_0$. Based on the results, it can be seen that under $H_0$, the type-I error rate of our test is well-controlled and close to the nominal level at each interim stage. Under $H_1$, the power of our test increases as $\delta$ increases, showing the consistency of our test procedure.

To further evaluate our method, we compare it with the classical two-sample t-test and the sequential test developed by Kharitonov et al. (2015). To apply the t-test, for each $T_k$, we apply the t-test to the

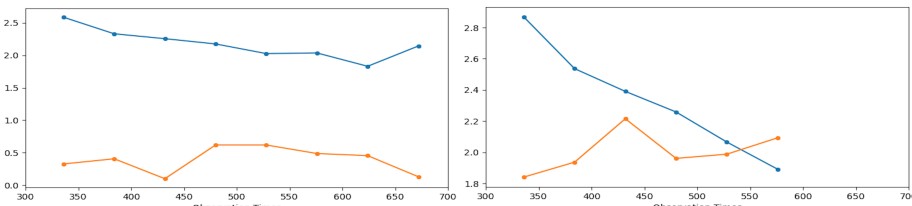

Figure 2: Our test statistic (the orange line) and the rejection boundary (the black line) in the A/A (left plot) and A/B (right plot) experiments.

data $\{A_t, Y_t\}_{0 \le t < T_k}$ and plot the corresponding empirical rejection probabilities in Figures 1(b) and 5(b) (Appendix G). Results for Kharitonov et al. (2015)'s test are reported in Figure 4 (Appendix G). Both competing methods fail to detect any carryover effects and do not have any power.

We next explain why several other methods mentioned in the introduction cannot be used for comparison. First, a lot of causal effects evaluation methods did not consider early termination. Consequently, they are unsuitable to apply in our numerical studies. Second, standard temporal difference learning method did not study the asymptotic distribution of the resulting value estimators. These results are critical for carrying out A/B testing. Finally, many methods proposed to use inverse propensity-score weighting. These methods are not valid for the alternating-time-interval design.

## 5.2 REAL DATA APPLICATION

We apply the proposed test to a real dataset from a ride-sharing platform. Order dispatching is one of the most critical problems in online ride-hailing platforms to adapt the operation and management strategy to the dynamics in demand and supply. The purpose of this study is to compare the performance of a newly developed strategy with a standard control strategy used in the platform. The new strategy is expected to reduce the answer time of passengers and increase drivers income. For a given order, the new strategy will dispatch it to a nearby driver that has not yet finished their previous ride request, but almost. In comparison, the standard control assigns orders to drivers that have completed their ride requests.

The experiment is conducted at a given city from December 3rd to December 16th. Dispatch strategies are executed based on alternating half-hourly time intervals. We also apply our test to a data from an A/A experiment (which compares the baseline strategy against itself), conducted from November 12th to November 25th. We expect that our test will not reject $H_0$ when applied to the data from the A/A experiment, since the two strategies used are essentially the same.

Both experiments last for two weeks. Thirty-minutes is defined as one time unit. We set $T_k = 48(k + 6)$ for $k = 1, \ldots, 8$. That is, the first interim analysis is performed at the end of the first week, followed by seven more at the end of each day during the second week. We choose the overall drivers' income in each time unit as the response. The new strategy is expected to reduce the answer time of passengers and increase drivers' income. Three time-varying variables are used to construct the state. The first two correspond to the number of requests (demand) and drivers' online time (supply) during each 30-minutes time interval. These factors are known to have large impact on drivers' income. The last one is the supply and demand equilibrium metric. This variable characterizes the degree that supply meets the demand and serves as an important mediator between past treatments and future outcomes.

To implement our test, we set $\gamma = 0.6$, $B = 1000$ and use a fourth-degree polynomial basis for $\Psi(\cdot)$, as in simulations. We use $\alpha_1(\cdot)$ as the spending function for interim analysis and set $\alpha = 0.05$. The test statistic and its corresponding rejection boundary at each interim stage are plotted in Figure 2. It can be seen that our test is able to conclude, at the end of the 12th day, that the new order dispatch strategy can significantly increase drivers' income, and meet more order requests. In addition, based on the dataset from the A/B experiment, we found that the new strategy reduces the answer time of orders by 2%, leading to almost 2% increment of drivers income. When applied to the data from the A/A experiment, we fail to reject $H_0$, as expected. For comparison, we also apply the two-sample t-test to the data collected from the A/B experiment. The corresponding p-value is 0.18. This result is consistent with our findings. Specifically, the treatment effect at a given time affects the distribution of drivers in the future, inducing interference in time. As shown in the toy example (see Section 4.1), the t-test cannot detect such carryover effects, leading to a low power. Our procedure, according to Theorem 2, has enough powers to discriminate $H_1$ from $H_0$.

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

**Input:** no. of basis functions $q$, no. of bootstrap samples $B$, an $\alpha$ spending function $\alpha(\cdot)$.
**Initialize:** $\mathcal{I} = \{1, \cdots, B\}$. **Set** $\widehat{\boldsymbol{\Omega}}, \widehat{\boldsymbol{\Sigma}}_0, \widehat{\boldsymbol{\Sigma}}_1$ to zero matrices, and $\widehat{\boldsymbol{\eta}}, \widehat{S}_1, \cdots, \widehat{S}_B$ to zero vectors.
**Compute $\boldsymbol{U}$** (see Section 4.2) using either Monte Carlo methods or numerical integration.
**For** $k = 1$ to $K$:
    **Step 1.** Online update of ATE.
    **For** $t = T_{k-1}$ to $T_k - 1$:
        $\widehat{\boldsymbol{\Sigma}}_a = (1 - t^{-1})\widehat{\boldsymbol{\Sigma}}_a + t^{-1}\xi(S_t, A_t)\{\xi(S_t, A_t) - \gamma\xi(S_{t+1}, a)\}^\top, a = 0, 1;$
        $\widehat{\boldsymbol{\eta}} = (1 - t^{-1})\widehat{\boldsymbol{\eta}} + t^{-1}\xi(S_t, A_t)Y_t.$
    **Set** $(\widehat{\beta}_{a,0}^\top, \widehat{\beta}_{a,1}^\top)^\top = \widehat{\boldsymbol{\Sigma}}_a^{-1}\widehat{\boldsymbol{\eta}}$ for $a \in \{0, 1\}$ and $\widehat{\tau} = \boldsymbol{U}^\top\widehat{\boldsymbol{\beta}}$.
    **Step 2.** Online update of the variance estimator.
    **Initialize** $\widehat{\boldsymbol{\Omega}}^*$ to a zero matrix.
    **For** $t = T_{k-1}$ to $T_k - 1$:
        $\widehat{\varepsilon}_{t,a} = Y_t + \gamma\Psi^\top(S_{t+1})\widehat{\beta}_{a,a} - \Psi^\top(S_t)\widehat{\beta}_{a,A_t}$ for $a = 0, 1;$
        $\widehat{\boldsymbol{\Omega}}^* = \widehat{\boldsymbol{\Omega}}^* + (\xi(S_t, A_t)^\top\widehat{\varepsilon}_{t,0}, \xi(S_t, A_t)^\top\widehat{\varepsilon}_{t,1})^\top(\xi(S_t, A_t)^\top\widehat{\varepsilon}_{t,0}, \xi(S_t, A_t)^\top\widehat{\varepsilon}_{t,1}).$
    **Set** $\widehat{\boldsymbol{\Sigma}}$ to a block diagonal matrix by aligning $\widehat{\boldsymbol{\Sigma}}_0$ and $\widehat{\boldsymbol{\Sigma}}_1$ along the diagonal of $\widehat{\boldsymbol{\Sigma}}$;
    **Set** $\widehat{\boldsymbol{\Omega}} = T_k^{-1}(T_{k-1}\widehat{\boldsymbol{\Omega}} + \widehat{\boldsymbol{\Omega}}^*)$ and the variance estimator $\widehat{\sigma}^2 = \boldsymbol{U}^\top\widehat{\boldsymbol{\Sigma}}^{-1}\widehat{\boldsymbol{\Omega}}\{\widehat{\boldsymbol{\Sigma}}^{-1}\}^\top\boldsymbol{U}$.
    **Step 3.** Bootstrap test statistic.
    **For** $b = 1$ to $B$:
        Generate $e_k^{(b)} \sim N(0, I_{4q})$; $\widehat{S}_b = \widehat{S}_b + \widehat{\boldsymbol{\Omega}}^{*1/2}e_k^{(b)}$; $\widehat{Z}_b^* = T_k^{-1/2}\widehat{\sigma}^{-1}\boldsymbol{U}^\top\widehat{\boldsymbol{\Sigma}}^{-1}\widehat{S}_b$;
    **Set** $z$ to be the upper $\{\alpha(t) - |\mathcal{I}^c|/B\}/(1 - |\mathcal{I}^c|/B)$-th percentile of $\{\widehat{Z}_b^*\}_{b\in\mathcal{I}}$.
    **Update** $\mathcal{I}$ and $\widehat{\boldsymbol{\Omega}}$ as $\mathcal{I} \leftarrow \{b \in \mathcal{I} : \widehat{Z}_b^* \leq z\}$;
    **Step 4.** Reject or not?
    **Reject** the null if $\sqrt{T_k}\widehat{\sigma}^{-1}\widehat{\tau} > z$.

**Algorithm 1:** The testing procedure

## A    MORE ON THE ALGORITHM

A pseudo algorithm summarizing our procedure is given in Algorithm 1. We next introduce some notations. The matrix $\widehat{\boldsymbol{\Omega}}(t)$ is defined by

$$\widehat{\boldsymbol{\Omega}}(t) = \frac{1}{t}\sum_{j=0}^{t-1}\begin{pmatrix} \xi_j\widehat{\varepsilon}_{j,0} \\ \xi_j\widehat{\varepsilon}_{j,1} \end{pmatrix}\begin{pmatrix} \xi_j\widehat{\varepsilon}_{j,0} \\ \xi_j\widehat{\varepsilon}_{j,1} \end{pmatrix}^\top, \tag{3}$$

where $\widehat{\varepsilon}_{j,0}$ and $\widehat{\varepsilon}_{j,1}$ are the temporal difference errors defined in Algorithm 1.

## B    EXTENSIONS

### B.1    EXTENSIONS TO DYNAMIC POLICIES

In this paper, we focus on comparing the long-term treatment effects between two nondynamic policies. The proposed method can be easily extended to handle dynamic policies as well. Specifically, consider two time-homogeneous policies $\pi_1$ and $\pi_2$ where each $\pi_j(s)$ measures the treatment assignment probability $\Pr(A_t = 1|S_t = s)$. Note that the integrated value difference function $\tau_0$ can be represented by

$$\int_s \{V(\pi_1; s) - V(\pi_2; s)\}\mathbb{G}(ds) = \int_s [\{Q(\pi_1; 1, s) - Q(\pi_1; 0, s)\}\pi_1(s)$$
$$-\{Q(\pi_2; 1, s) - Q(\pi_2; 0, s)\}\pi_2(s) + Q(\pi_1; 0, s) - Q(\pi_2; 0, s)]\mathbb{G}(ds).$$

The Q-estimators can be similarly computed via temporal difference learning. More specifically, for a given policy $\pi$, let

$$\widehat{Q}_t(\pi; a, s) = \Psi^\top(s)\widehat{\boldsymbol{\Sigma}}_\pi^{-1}(t)\left\{\frac{1}{t}\sum_{j<t}\xi(S_j, A_j)Y_j\begin{pmatrix} A_j \\ 1 - A_j \end{pmatrix}\right\}, \tag{4}$$

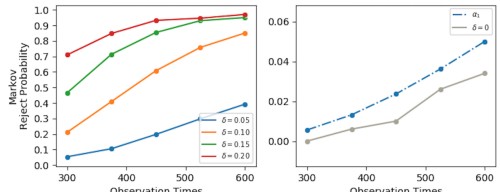 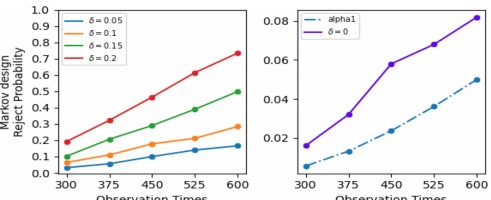

(a) The proposed test under $H_1$ (left) and $H_0$ (right)  (b) DRL-based test under $H_1$ (left) and $H_0$ (right)

Figure 3: Empirical rejection probabilities of our test and the DRL-based test.

be the Q-estimator given the data $\{(S_j, A_j, Y_j)\}_{j<t}$ where $\widehat{\Sigma}_\pi(t) = t^{-1} \sum_{j<t} \Sigma_j$ where $\Sigma_j$ is defined by

$$\left[ \begin{array}{cc} \Psi(S_j)(1-A_j)\{\Psi(S_j) - \gamma\Psi(S_{j+1})(1-\pi(S_{j+1}))\}^\top & -\gamma\Psi(S_j)(1-A_j)\Psi^\top(S_{j+1})\pi(S_{j+1}) \\ -\gamma\Psi(S_j)A_j\Psi^\top(S_{j+1})\pi(S_{j+1}) & \Psi(S_j)A_j\{\Psi(S_j) - \gamma\Psi(S_{j+1})(1-\pi(S_{j+1}))\}^\top \end{array} \right].$$

We can plug-in the Q-estimator in equation 4 to estimate $\tau_0$. The corresponding variance estimator and the resulting test statistic can be similarly derived. A bootstrap procedure can be similarly developed as in Section 4.3 for sequential testing. We omit the details for brevity.

### B.2 EXTENSIONS TO OTHER NONPARAMETRIC ESTIMATORS

In addition to temporal difference learning, other existing OPE methods could be potentially coupled with the proposed bootstrap procedure for online sequential testing. We use the double reinforcement learning method (DRL, Kallus & Uehara, 2019) as an example.

First, we remark that DRL requires the system to be ergodic and use an inverse propensity-score weighted method to construct the value estimator. As such, it might not be applicable to the alternating-time-interval design and the adaptive design.

Second, in the Markov design, it could be coupled with our bootstrap procedure for sequential testing. We compare such a procedure with our proposed method using the simulation setting in Section 5.1, and report the rejection probabilities in Figure 3. It can be seen that DRL-based test has some inflated type-I errors under $H_0$ and is less powerful than our procedure under $H_1$.

We next outline the procedure. Specifically, at the $k$th interim stage, we compute the test statistic

$$Z_k = \frac{1}{\sqrt{T_k}\widehat{\sigma}_k} \left\{ \sqrt{T}_{k-1}\widehat{\sigma}_{k-1}Z_{k-1} + \sum_{t=T_{k-1}}^{T_k-1} \psi_t \right\},$$

where

$$\psi_t = \int_s \{\widehat{Q}_k(1;1,s) - \widehat{Q}_k(0;0,s)\}\mathbb{G}ds + \gamma\frac{A_t}{b(S_t)}\widehat{\omega}_k(1;S_t)\{R_t + \widehat{Q}_k(1;1,S_{t+1}) - \widehat{Q}_k(1;A_t,S_t)\}$$

$$-\gamma\frac{1-A_t}{1-b(S_t)}\widehat{\omega}_k(0;S_t)\{R_t + \widehat{Q}_k(0;0,S_{t+1}) - \widehat{Q}_k(0;A_t,S_t)\},$$

$\widehat{Q}_k$ and $\widehat{\omega}_k$ denote the estimated Q- and marginal density ratio functions based on the data collected at the $k$th stage, and

$$\widehat{\sigma}_k^2 = \frac{T_{k-1}\widehat{\sigma}_{k-1}^2 + \sum_{t=T_{k-1}}^{T_k-1} \psi_t^2}{T_k}.$$

The bootstrapped sample can be constructed as

$$Z_k^* = \sqrt{T_{k-1}}\widehat{\sigma}_k^{-1}\widehat{\sigma}_{k-1}Z_{k-1}^* + \sqrt{T_k - T_{k-1}}N(0,1).$$

Then similar to Algorithm 1, we can decide whether to reject $H_0$ or not based on the test statistic $Z_k$ and the bootstrap samples. This algorithm can be implemented online provided that $\widehat{Q}$ and $\widehat{\omega}$ can be computed online.

## C    MORE ON THE WILD BOOTSTRAP ALGORITHM

We first provide an example to show that our test statistics do not have the canonical joint distribution. This motivates us to propose a wild bootstrap algorithm. We next present some details on the bootstrap algorithm.

Let $\{Z_k\}_k$ be the sequence of test statistics conducted at each interim stage. These test statistics are said to have canonical joint distribution with information levels $\{\mathcal{I}_k\}_k$ for the parameter $\theta$ if:

(i) $(Z_1, Z_2, \cdots, Z_K)^\top$ is asymptotically normal,

(ii) $\mathrm{E}Z_k = \theta\sqrt{\mathcal{I}_k} + o(1)$,

(iii) $\mathrm{Cov}(Z_{k_1}, Z_{k_2}) = \sqrt{\mathcal{I}_{k_1}/\mathcal{I}_{k_2}} + o(1)$.

See also Equation (3.1) in Jennison & Turnbull (1999).

Unlike the settings where observations are independent across time, (iii) is likely to be violated in our setup when adaptive design is used. This is due to the existence of carryover effects in time. Specifically, when treatment effects are adaptively generated, the behavior policy at difference stages are likely to vary. Due to the carryover effects in time, the state vectors at difference stages have different distribution functions.

According to Part 3 of the proof of Theorem 3, we have for any $k_1 \leq k_2$ that

$$\mathrm{Cov}(Z_{k_1}, Z_{k_2}) = \sqrt{\frac{T_{k_1}}{T_{k_2}}} \times \underbrace{\boldsymbol{U}^\top \boldsymbol{\Sigma}^{-1}(T_{k_1})\boldsymbol{\Omega}(T_{k_1})\{\boldsymbol{\Sigma}^{-1}(T_{k_2})\}^\top \boldsymbol{U}}_{\eta_{k_1,k_2}} + o(1).$$

The matrices $\boldsymbol{\Sigma}(k)$ and $\boldsymbol{\Omega}(k)$ depend the distributions of state vectors and are likely to differ for different $k$. Consequently, the second term $\eta_{k_1,k_2}$ on the right-hand-side depends on both $k_1$ and $k_2$. As such, (iii) is violated.

The idea of our bootstrap algorithm is to generate bootstrap samples $\{\widehat{Z}^{\mathrm{MB}}(t)\}_t$ that have asymptotically the same joint distribution as $\{\sqrt{t}\widehat{\sigma}^{-1}(t)(\widehat{\tau}(t) - \tau_0)\}_t$. Specifically, let $\{\zeta_t\}_{t \geq 0}$ be a sequence of i.i.d. random variables independent of the observed data. Define

$$\widehat{\boldsymbol{\beta}}^{\mathrm{MB}}(t) = \widehat{\boldsymbol{\Sigma}}^{-1}(t)\left\{\frac{1}{t}\sum_{j<t}\xi(S_j, A_j)\zeta_j\left(\begin{array}{c}\widehat{\varepsilon}_{j,0}\\\widehat{\varepsilon}_{j,1}\end{array}\right)\right\}, \tag{5}$$

where $\widehat{\varepsilon}_{t,a}$ is the temporal difference error defined in Algorithm 1. Based on $\widehat{\boldsymbol{\beta}}^{\mathrm{MB}}(t)$, one can define the bootstrap sample $\widehat{Z}^{\mathrm{MB}}(t) = \sqrt{t}\widehat{\sigma}^{-1}(t)\boldsymbol{U}^\top\widehat{\boldsymbol{\beta}}^{\mathrm{MB}}(t)$.

We remark that although the wild bootstrap method is developed under the i.i.d. settings, it is valid under our setup as well. This is due to that under CMIA, $\widehat{\boldsymbol{\beta}}(t) - \boldsymbol{\beta}^*$ forms a martingale sequence with respect to the filtration $\{(S_j, A_j, Y_j) : j < t\}$. This guarantees that the covariance matrices of $\widehat{\boldsymbol{\beta}}^{\mathrm{MB}}(t)$ and $\widehat{\boldsymbol{\beta}}(t)$ are asymptotically equivalent. As such, the bootstrap approximation is valid.

However, calculating $\widehat{\boldsymbol{\beta}}^{\mathrm{MB}}(T_k)$ requires $O(T_k)$ operations. The time complexity of the resulting bootstrap algorithm is $O(BT_k)$ up to the $k$-th interim stage, where $B$ is the total number of bootstrap samples. This can be time consuming when $\{T_k - T_{k-1}\}_{k=1}^K$ are large.

## D    MORE ON THE DESIGNS

In D3, we require $b^{(k)}$ to be strictly bounded between 0 and 1. Suppose an $\epsilon$-greedy policy is used, i.e. $b^{(k)}(s) = \epsilon/2 + (1 - \epsilon)\widehat{\pi}^{(k)}(s)$, where $\widehat{\pi}^{(k)}$ denotes some estimated optimal policy. It follows that $\epsilon/2 \leq b^{(k)}(s) \leq 1 - \epsilon/2$ for any $s$. Such a requirement is automatically satisfied.

For any behaviour policy $b$ in D1-D3, define $S_t^*(\bar{b}_{t-1})$ and $Y_t^*(\bar{b}_t)$ as the potential outcomes at time $t$, where $\bar{b}_t$ denotes the action history assigned according to $b$. When $b$ is a random policy as in D1 or D3, definitions of these potential outcomes are more complicated than those under a

deterministic policy (see Luckett et al., 2019). When $b$ is a stationary policy, it follows from MA that $\{S_{t+1}^*(\bar{b}_t)\}_{t\geq -1}$ forms a time-homogeneous Markov chain. When $b$ follows the alternating-time-interval design, both $\{S_{2t}^*(\bar{b}_{2t-1})\}_{t\geq 0}$ and $\{S_{2t+1}^*(\bar{b}_{2t})\}_{t\geq 0}$ form time-homogeneous Markov chains.

We next show that $(Z_1, \cdots, Z_K)$ is asymptotically multivariate normal and provide a consistent covariance estimator.

**Theorem 3 (Limiting distributions)** *Assume C1-C3 hold. Assume all immediate rewards are uniformly bounded variables, the density function of $S_0$ is uniformly bounded on $\mathbb{S}$ and $q$ satisfies $q = o(\sqrt{T}/\log T)$. Then under either D1, D2 or D3, we have*
- *$\{Z_k\}_{1\leq k\leq K}$ are jointly asymptotically normal;*
- *their asymptotic means are non-positive under $H_0$;*
- *their covariance matrix can be consistently estimated by some $\widehat{\boldsymbol{\Xi}}$, whose $(k_1, k_2)$-th element $\widehat{\Xi}_{k_1,k_2}$ equals $\sqrt{T_{k_1}/T_{k_2}}\boldsymbol{U}^\top\widehat{\boldsymbol{\Sigma}}^{-1}(T_{k_1})\widehat{\boldsymbol{\Omega}}(T_{k_1})\{\widehat{\boldsymbol{\Sigma}}^{-1}(T_{k_2})\}^\top\boldsymbol{U}/\{\widehat{\sigma}(T_{k_1})\widehat{\sigma}(T_{k_2})\}$.*

## E  TECHNICAL CONDITIONS

To simplify the presentation, we assume all state variables are continuous. The immediate reward and the density function of $S_0$ are bounded.

### E.1  CONDITION C1

C1 Suppose (i) holds. Assume (ii) holds under D1, (iii) holds under D2 and (ii), (iv) hold under D3.
(i) The transition kernel $\mathcal{P}$ is absolutely continuous and satisfies $\mathcal{P}(ds; a, s') = p(s; a, s')ds$ for some transition density function $p$. In addition, assume $p$ is uniformly bounded away from 0 and $\infty$.
(ii) The Markov chain $\{S_t^*(\bar{b}_{t-1}^{(0)})\}_{t\geq 0}$ formed under the behaviour policy $b^{(0)}$ is geometrically ergodic, i.e. there exists some function $M$ on $\mathbb{S}$, some constant $0 \leq \rho < 1$ and some probability density function $\Pi$ such that $\int_{s\in\mathbb{S}} M(s)\Pi(ds) < +\infty$ and

$$\left\|\Pr(S_t^*(\bar{b}_{t-1}^{(0)}) \in \mathcal{S}|S_0 = s) - \Pi(\mathcal{S})\right\|_{TV} \leq M(s)\rho^t, \quad \forall t \geq 0, s \in \mathbb{S}, \mathcal{S} \subseteq \mathbb{S},$$

where $\|\cdot\|_{TV}$ denotes the total variation norm.
(iii) The Markov chains $\{S_{2t}^*(\bar{b}_{2t})\}_{t\geq 0}$ and $\{S_{2t+1}^*(\bar{b}_{2t+1})\}_{t\geq 0}$ are geometrically ergodic.
(iv) For any $k = 1, \cdots, K-1$, the following events occur with probability tending to 1: the Markov chain $\{S_t^*(\bar{b}_{t-1}^{(k)})\}_{t\geq 0}$ is geometrically ergodic; $\sup_{s\in\mathbb{S}} |b^{(k)}(s) - b^*(s)| \overset{P}{\to} 0$ for some $b^*(\cdot)$; the stationary distribution of $\{S_t^*(\bar{b}_{t-1}^{(k)})\}_{t\geq 0}$ will converge to some $\Pi^*$ in total variation.

**Remark**: By C1(ii), $\Pi$ is the stationary distribution of $\{S_t^*(\bar{b}_{t-1}^{(0)})\}_{t\geq 0}$. It follows that

$$\Pi(\mathcal{S}) = \sum_{a\in\{0,1\}} \int_{s\in\mathbb{S}} \mathcal{P}(\mathcal{S}; a, s)\{ab^{(1)}(s) + (1-a)b^{(0)}(s)\}\Pi(ds),$$

for any $\mathcal{S} \subseteq \mathbb{S}$. By C1(i), we obtain

$$\Pi(\mathcal{S}) = \sum_{a\in\{0,1\}} \int_{s\in\mathbb{S}} \int_{s'\in\mathcal{S}} [a\{1 - b^{(0)}(s)\} + (1-a)b^{(0)}(s)]p(s'; a, s)ds'\Pi(ds)$$

$$= \int_{s'\in\mathcal{S}} \underbrace{\sum_{a\in\{0,1\}} \int_{s\in\mathbb{S}} [a\{1 - b^{(0)}(s)\} + (1-a)b^{(0)}(s)]p(s'; a, s)\Pi(ds)}_{\mu(s')} \, ds'. \tag{6}$$

This implies that $\mu(\cdot)$ is the density function of $\Pi$. Since $p$ is uniformly bounded away from 0 and $\infty$, so is $\mu$.

Under C1(iv), for any $k \in \{1, \cdots, K-1\}$, there exist some $M^{(k)}(\cdot), \Pi^{(k)}(\cdot)$ and $\rho^{(k)}$ that satisfy $\int_{s\in\mathbb{S}} M^{(k)}(s)\Pi^{(k)}(ds) < +\infty$ and

$$\left\|\Pr(S_t^*(\bar{b}_{t-1}^{(k)}) \in \mathcal{S}|S_0 = s) - \Pi^{(k)}(\mathcal{S})\right\|_{TV} \leq M^{(k)}(s)\{\rho^{(k)}\}^t, \quad \forall t \geq 0, s \in \mathbb{S}, \mathcal{S} \subseteq \mathbb{S}, \tag{7}$$

with probability tending to 1. Since $b^{(k)}$ is a function of the observe data history, so are $M^{(k)}(\cdot)$, $\Pi^{(k)}(\cdot)$ and $\rho^{(k)}$.

Suppose an $\epsilon$-greedy policy is used, i.e. $b^{(k)}(s) = \epsilon/2 + (1 - \epsilon)\widehat{\pi}^{(k)}(s)$ where $\widehat{\pi}^{(k)}$ denotes some estimated optimal policy. Then the condition $\sup_{s\in\mathbb{S}} |b^{(k)}(s) - b^*(s)| \xrightarrow{P} 0$ requires $\widehat{\pi}^{(k)}$ to converge. The total variation distance between the one-step transition kernel under $\bar{b}^{(k)}$ and that under $b^*$ can be bounded by

$$\sup_s |\Pr(S_1^*(b^{(k)}) \in \mathcal{S}|S_0 = s) - \Pr(S_1^*(b^*) \in \mathcal{S}|S_0 = s)| \leq \sup_s |b^{(k)}(s) - b^*(s)| \sup_{s,s',a} p(s'; a, s),$$

and converges to zero in probability. When the markov chain $\{S_t^*(\bar{b}_{t-1}^{(k)})\}_{t\geq 0}$ is uniformly ergodic, it follows from Theorems 2 and 3 of Rabta & Aïssani (2018) that $\|\Pi^{(k)} - \Pi^*\|_{TV} \to 0$ where $\Pi^*$ corresponds to the stationary distribution of $\{S_t^*(\bar{b}_{t-1}^*)\}$. The last condition in C1(iv) is thus satisfied.

## E.2 CONDITION C2

C2(i) Assume there exists some $\beta^*$ such that

$$\sup_{a',a\in\{0,1\},s\in\mathbb{S}} |Q(a'; a, s) - \Psi^\top(s)\beta_{a',a}^*| = o(T^{-1/2}).$$

(ii) Assume there exists some constant $\bar{c}^* \geq 1$ such that

$$(\bar{c}^*)^{-1} \leq \lambda_{\min}\left\{\int_{s\in\mathbb{S}} \Psi(s)\Psi^\top(s)ds\right\} \leq \lambda_{\max}\left\{\int_{s\in\mathbb{S}} \Psi(s)\Psi^\top(s)ds\right\} \leq \bar{c}^*, \tag{8}$$

and $\sup_s \|\Psi(s)\|_2 = O(\sqrt{q})$.
(iii) Assume $\liminf_q \|\int_{s\in\mathbb{S}} \Psi(s)\mathbb{G}(ds)\|_2 > 0$.

**Remark**: For any $a, a' \in \{0, 1\}$, suppose $Q(a'; a, s)$ is $p$-smooth as a function of $s$ (see e.g. Stone, 1982, for the definition of $p$-smoothness). When tensor product B-splines or wavelet basis functions (see Section 6 of Chen & Christensen, 2015, for an overview of these bases) are used for $\Psi(\cdot)$, we have

$$\sup_{a',a\in\{0,1\},s\in\mathbb{S}} |Q(a'; a, s) - \Psi^\top(s)\beta_{a',a}^*| = o(q^{-p/d}),$$

under certain mild conditions. See Section 2.2 of Huang (1998) for details. It follows that Condition C2(i) automatically holds when the number of basis functions $q$ satisfies $q \gg T^{d/(2p)}$.

Condition C2(ii) is satisfied when tensor product B-splines or wavelet basis is used. For B-spline basis, the assertion in equation 8 follows from the arguments used in the proof of Theorem 3.3, Burman & Chen (1989). For wavelet basis, the assertion in equation 8 follows from the arguments used in the proof of Theorem 5.1, Chen & Christensen (2015). For both bases, the number of nonzero elements in $\Psi(\cdot)$ is bounded by some constant. Moreover, each basis function is uniformly bounded by $O(\sqrt{q})$. The condition $\sup_s \|\Psi(s)\|_2 = O(\sqrt{q})$ thus holds.

Condition C2(iii) automatically holds for tensor product B-splines basis. Notice that $\mathbf{1}^\top \Psi(s) = q^{1/2}$ for any $s \in \mathbb{S}$ where $\mathbf{1}$ denotes a vector of ones. It follows from Cauchy-Schwarz inequality that

$$\sqrt{q}\left\|\int_{s\in\mathbb{S}} \Psi(s)\mathbb{G}(ds)\right\|_2 \geq \left\|\int_{s\in\mathbb{S}} \mathbf{1}^\top \Psi(s)\mathbb{G}(ds)\right\|_2 = \sqrt{q}.$$

C2(iii) is thus satisfied.

## E.3 CONDITION C3

Let $\varepsilon^*(a', a) = Y_0^*(a) + \gamma Q(a'; a, S_1^*(a)) - Q(a'; a, S_0)$.

C3 Assume $\inf_q \inf_{a',a\in\{0,1\},s\in\mathbb{S}} \text{Var}\{\varepsilon^*(a', a)|S_0 = s\} > 0$ and $\sup_q \sup_{a\in\{0,1\},s\in\mathbb{S}} \rho_\varepsilon(a, s) < 1$ where

$$\rho_\varepsilon(a, s) = \frac{\text{E}\{\varepsilon^*(0, a)\varepsilon^*(1, a)|S_0 = s\}}{\sqrt{\text{Var}(\varepsilon^*(0, a)|S_0 = s)\text{Var}(\varepsilon^*(1, a)|S_0 = s)}}.$$

Here, $\rho_\varepsilon$ corresponds to the partial correlation of $\varepsilon^*(0, a)$ and $\varepsilon^*(1, a)$ given $S_0$.

# F    TECHNICAL PROOFS

## F.1    PROOF OF LEMMA 1

To prove Lemma 1, we state the following lemma.

**Lemma 2** *Under MA and CMIA, $Q(a'; a, s) = r(a, s) + \gamma \int_{s'} Q(a'; a', s') \mathcal{P}(ds'; a, s)$ for any $(s, a)$.*

*Proof of Lemma 2:* For any $a, a' \in \{0, 1\}$, define the potential outcome $Y_t^*(a', a)$ and $S_t^*(a', a)$ as the reward and state variables that would occur at time $t$ had the agent assigned Treatment $a$ at the initial time point and Treatment $a'$ afterwards.

Let $\mathcal{P}_{a'}^t(\mathbb{S}, a, s) = \Pr\{S_t^*(a', a) \in \mathbb{S} | S_0 = s\}$ for any $\mathbb{S} \subseteq \mathbb{S}, a, a' \in \{0, 1\}, s \in \mathbb{S}$ and $t \geq 0$. We break the proof into two parts. In Part 1, we show Lemma 2 holds when the following is satisfied:

$$\Pr\{S_{t+1}^*(a', a) \in \mathbb{S} | S_1^*(a) = s, S_0\} = \mathcal{P}_{a'}^t(\mathbb{S}, a', s), \tag{9}$$

In Part2, we show equation 9 holds.

*Part 1:* Under CMIA, we have

$$\begin{aligned} \mathrm{E}\{Y_t^*(a', a) | S_0 = s\} &= \mathrm{E}[\mathrm{E}\{Y_t^*(a', a) | S_t^*(a', a), S_0 = s\} | S_0 = s] \\ &= \mathrm{E}\{r(\pi(S_t^*(a', a)), S_t^*(a', a)) | S_0 = s\}. \end{aligned} \tag{10}$$

It follows that

$$Q(a'; a, s) = \sum_{t \geq 0} \gamma^t \mathrm{E}\{r(\pi(S_t^*(a', a)), S_t^*(a', a)) | S_0 = s\}. \tag{11}$$

Similar to equation 10, we can show

$$\begin{aligned} \mathrm{E}\{Y_{t+1}^*(a', a) | S_0 = s\} &= \mathrm{E}\{r(\pi(S_{t+1}^*(a', a)), S_{t+1}^*(a', a)) | S_0 = s\} \\ &= \mathrm{E}[\mathrm{E}\{r(\pi(S_{t+1}^*(a', a)), S_{t+1}^*(a', a)) | S_1^*(a), S_0 = s\} | S_0 = s], \end{aligned}$$

and hence

$$\sum_{t \geq 0} \gamma^t \mathrm{E}\{Y_{t+1}^*(a', a) | S_0 = s\} = \mathrm{E}\left[\sum_{t \geq 0} \gamma^t \, \mathrm{E}\{r(\pi(S_{t+1}^*(a', a)), S_{t+1}^*(a', a)) \big| S_1^*(a), S_0 = s\} | S_0 = s\right].$$

By equation 9, the conditional distribution of $S_{t+1}^*(a', a)$ given $S_1^*(a) = s$ and $S_0$ are the same as the conditional distribution of $S_t^*(a', a)$ given $S_0 = s$. It follows that from equation 11 that

$$\sum_{t \geq 0} \gamma^t \mathrm{E}\{Y_{t+1}^*(a', a) | S_0 = s\} = \mathrm{E}\{Q(a'; a, S_1^*(a)) | S_0 = s\}.$$

This together with the definition of Q function and CMIA yields

$$Q(a'; a, s) = r(a, s) + \gamma \left[\sum_{t \geq 0} \gamma^t \mathrm{E}\{Y_{t+1}^*(a', a) | S_0 = s\}\right] = r(a, s) + \gamma \mathrm{E}\{Q(a'; a, S_1^*(a)) | S_0 = s\}. \tag{12}$$

Under MA, we have

$$\mathrm{E}\{Q(a'; a, S_1^*(a)) | S_0 = s\} = \int_{s' \in \mathbb{S}} Q(a'; a, s') \mathcal{P}(ds'; a, s).$$

Combining this together with equation 12 yields the desired result.

*Part 2:* We use induction to prove equation 9. When $t = 0$, it trivially holds.

Suppose equation 9 holds for $t = k$. In the following, we show equation 9 holds for $t = k + 1$. Under MA, we have

$$\begin{aligned} \Pr\{S_{k+2}^*(a', a) \in \mathbb{S} | S_1^*(a) = s, S_0\} &= \mathrm{E}[\Pr\{S_{k+2}^*(a', a) \in \mathbb{S} | S_{k+1}^*(a', a), S_1^*(a) = s, S_0\} | S_1^*(a) = s, S_0] \\ &= \mathrm{E}[\mathcal{P}(\mathbb{S}; a', S_{k+1}^*(a', a)) | S_1^*(a) = s, S_0]. \end{aligned}$$

Since we have shown equation 9 holds for $t = k$, it follows that

$$\Pr\{S^*_{k+2}(a', a) \in \mathbb{S} | S^*_1(a) = s, S_0\} = \int_{s' \in \mathbb{S}} \mathcal{P}(\mathbb{S}; a', s') \mathcal{P}^k_{a'}(ds', a', s).$$

Similarly, we can show

$$\mathcal{P}^{k+1}_{a'}(\mathbb{S}, a', s) = \Pr\{S^*_{k+1}(a', a') \in \mathbb{S} | S_0 = s\} = \int_{s' \in \mathbb{S}} \mathcal{P}(\mathbb{S}; a', s') \mathcal{P}^k_{a'}(ds', a', s).$$

The proof is hence completed.

*Proof of Lemma 1*: By CA, it is equivalent to show

$$\mathrm{E}\{Q(a'; A_t, S^*_t(\bar{A}_{t-1})) - Y^*_t(\bar{A}_t) - \gamma Q(a'; a', S^*_{t+1}(\bar{A}_t))\} \varphi(A_t, S^*_t(\bar{A}_{t-1})) = 0.$$

Let $\mathbb{S}_0$ denote the support of $S_0$. For any $s_0 \in \mathbb{S}_0$, it suffices to show

$$\mathrm{E}\{Q(a'; A_t, S^*_t(\bar{A}_{t-1})) - Y^*_t(\bar{A}_t) - \gamma Q(a'; a', S^*_{t+1}(\bar{A}_t)) \varphi(A_t, S^*_t(\bar{A}_{t-1})) | S_0 = s_0\} = 0.$$

This is equivalent to show

$$\mathrm{E}\{Q(a'; A_t, S^*_t(\bar{A}_{t-1})) - Y^*_t(\bar{A}_t) - \gamma Q(a'; a', S^*_{t+1}(\bar{A}_t)) \varphi(A_t, S^*_t(\bar{A}_{t-1})) \mathbb{I}(A_0 = a_0)\} | S_0 = s_0] = 0,$$

for any $s_0 \in \mathbb{S}_0, a_0 \in \{0, 1\}$.

Let $\mathcal{A}_0(s_0) = \{a \in \{0, 1\} : \Pr(A_0 = a | S_0 = s) > 0\}$. It suffices to show for any $s_0 \in \mathbb{S}_0, a_0 \in \mathcal{A}_0(s_0)$,

$$\mathrm{E}\{Q(a'; A_t, S^*_t(\bar{A}_{t-1})) - Y^*_t(\bar{A}_t) - \gamma Q(a'; a', S^*_{t+1}(\bar{A}_t)) \varphi(A_t, S^*_t(\bar{A}_{t-1})) \mathbb{I}(A_0 = a_0) | S_0 = s_0\} = 0,$$

or equivalently,

$$\mathrm{E}\{Q(a'; A_t, S^*_t(\bar{A}_{t-1})) - Y^*_t(\bar{A}_t) - \gamma Q(a'; a', S^*_{t+1}(\bar{A}_t)) \varphi(A_t, S^*_t(\bar{A}_{t-1})) | S_0 = s_0, A_0 = a_0\} = 0. \quad (13)$$

Let $\bar{s}_j = (s_0, s_1, \cdots, s_j)^\top$, $\bar{y}_j = (y_0, y_1, \cdots, y_j)^\top$, $\bar{S}_j = (S_0, S_1, \cdots, S_j)^\top$ and $\bar{Y}_j = (Y_0, Y_1, \cdots, Y_j)^\top$. We can recursively define the sets $\mathcal{Y}_j(\bar{s}_j, \bar{a}_j, \bar{y}_{j-1})$, $\mathbb{S}_{j+1}(\bar{s}_j, \bar{a}_j, \bar{y}_j)$, $\mathcal{A}_{j+1}(\bar{s}_{j+1}, \bar{a}_j, \bar{y}_j)$ to be the supports of $Y_j, S_{j+1}, A_{j+1}$ conditional on $(\bar{S}_j = \bar{s}_j, \bar{A}_j = \bar{a}_j, \bar{Y}_{j-1} = \bar{y}_{j-1})$, $(\bar{S}_j = \bar{s}_j, \bar{A}_j = \bar{a}_j, \bar{Y}_j = \bar{y}_j)$, $(\bar{S}_{j+1} = \bar{s}_{j+1}, \bar{A}_j = \bar{a}_j, \bar{Y}_j = \bar{y}_j)$ respectively, for $j \geq 0$. Similar to equation 13, it suffices to show

$$\mathrm{E}\{Q(a'; A_t, S^*_t(\bar{A}_{t-1})) - Y^*_t(\bar{A}_t) - \gamma Q(a'; a', S^*_{t+1}(\bar{A}_t)) \varphi(A_t, S^*_t(\bar{A}_{t-1})) | \bar{S}_t = \bar{s}_t, \bar{A}_t = \bar{a}_t, \bar{Y}_{t-1} = \bar{y}_{t-1}\} = 0,$$

for any $s_0 \in \mathbb{S}_0, a_0 \in \mathcal{A}_0(s_0), y_0 \in \mathcal{Y}_0(s_0, a_0), \cdots, s_t \in \mathbb{S}_t(\bar{s}_{t-1}, \bar{a}_{t-1}, \bar{y}_{t-1}), a_t \in \mathcal{A}_t(\bar{s}_t, \bar{a}_{t-1}, \bar{y}_{t-1})$. This is equivalent to show

$$\mathrm{E}\{Q(a'; a_t, S^*_t(\bar{a}_{t-1})) - Y^*_t(\bar{a}_t) - \gamma Q(a'; a', S^*_{t+1}(\bar{a}_t)) | \bar{S}_t = \bar{s}_t, \bar{A}_t = \bar{a}_t, \bar{Y}_{t-1} = \bar{y}_{t-1}\} = 0. \quad (14)$$

By construction, we have $\Pr(A_t = a_t | \bar{S}_t = \bar{s}_t, \bar{Y}_{t-1} = \bar{y}_{t-1}, \bar{A}_{t-1} = \bar{a}_{t-1}) > 0$. Under SRA, the left-hand-side (LHS) of equation 14 equals

$$\mathrm{E}\{Q(a'; a_t, S^*_t(\bar{a}_{t-1})) - Y^*_t(\bar{a}_t) - \gamma Q(a'; a', S^*_{t+1}(\bar{a}_t)) | \bar{S}_t = \bar{s}_t, \bar{A}_{t-1} = \bar{a}_{t-1}, \bar{Y}_{t-1} = \bar{y}_{t-1}\}. \quad (15)$$

Notice that the conditioning event is the same as $\{S^*_t(\bar{a}_{t-1}) = s_t, Y^*_{t-1}(\bar{a}_{t-1}) = y_{t-1}, \bar{S}_{t-1} = \bar{s}_{t-1}, \bar{A}_{t-1} = \bar{a}_{t-1}, \bar{Y}_{t-2} = \bar{y}_{t-2}\}$. Under SRA, equation 15 equals

$$\mathrm{E}\{Q(a'; a_t, S^*_t(\bar{a}_{t-1})) - Y^*_t(\bar{a}_t) - \gamma Q(a'; a', S^*_{t+1}(\bar{a}_t)) | S^*_t(\bar{a}_{t-1}) = s_t, Y^*_{t-1}(\bar{a}_{t-1}) = y_{t-1},$$
$$\bar{S}_{t-1} = \bar{s}_{t-1}, \bar{A}_{t-2} = \bar{a}_{t-2}, \bar{Y}_{t-2} = \bar{y}_{t-2}\}.$$

By recuisvely applying SRA, we can show the left-hand-side of equation 14 equals

$$\mathrm{E}[Q(a'; a_t, S^*_t(\bar{a}_{t-1})) - Y^*_t(\bar{a}_t) - \gamma Q(a'; a', S^*_{t+1}(\bar{a}_t)) | \{S^*_j(\bar{a}_{j-1}) = s_j\}_{1 \leq j \leq t}, \{Y^*_j(\bar{a}_j) = y_j\}_{1 \leq j \leq t-1}].$$

This is equal to zero by MA, CMIA and Lemma 2. The proof is hence completed.

### F.2 PROOF OF THEOREM 3

#### F.2.1 PROOF UNDER D1

We begin by providing an outline of the proof. The proof is divided into three steps. In the first step, we show for any $T_1 \le t \le T_k$, the estimator $\widehat{\boldsymbol{\beta}}(t)$ satisfies the following linear representation,

$$\widehat{\boldsymbol{\beta}}(t) - \boldsymbol{\beta}^* = \boldsymbol{\Sigma}^{-1}(t) \underbrace{\left\{ \frac{1}{t} \sum_{j=0}^{t-1} \left( \begin{array}{c} \xi_j \varepsilon_{j,0} \\ \xi_j \varepsilon_{j,1} \end{array} \right) \right\}}_{\zeta_1(t)} + o_p(t^{-1/2}), \tag{16}$$

where $\boldsymbol{\Sigma}(t) = E\widehat{\boldsymbol{\Sigma}}(t)$ and $\varepsilon_{j,a} = Y_j + \gamma Q(a; a, S_{j+1}) - Q(a; A_j, S_j)$ for $a = 0, 1$. Based on this representation, in the second step, we show the asymptotic normality of $\widehat{\tau}(t)$. Specifically, we show

$$\frac{\sqrt{t}\{\widehat{\tau}(t) - \tau_0\}}{\widehat{\sigma}(t)} \xrightarrow{d} N(0, 1).$$

In the last step, we prove Theorem 3.

**Part 1:** By definition, we have

$$
\begin{aligned}
\widehat{\boldsymbol{\beta}}(t) - \boldsymbol{\beta}^* &= \widehat{\boldsymbol{\Sigma}}^{-1}(t) \left\{ \frac{1}{t} \sum_{j=0}^{t-1} \left( \begin{array}{c} \xi_j Y_j \\ \xi_j Y_j \end{array} \right) - \widehat{\boldsymbol{\Sigma}}(t) \boldsymbol{\beta}^* \right\} = \widehat{\boldsymbol{\Sigma}}^{-1}(t) \left[ \frac{1}{t} \sum_{j=0}^{t-1} \left\{ \left( \begin{array}{c} \xi_j Y_j \\ \xi_j Y_j \end{array} \right) - \boldsymbol{\Sigma}_j \boldsymbol{\beta}^* \right\} \right] \\
&= \widehat{\boldsymbol{\Sigma}}^{-1}(t) \left[ \frac{1}{t} \sum_{j=0}^{t-1} \left\{ \begin{array}{c} \xi_j \{Y_j - \Psi^\top(S_j) \beta_{0,A_j}^* + \gamma \Psi^\top(S_{j+1}) \beta_{0,0}^*\} \\ \xi_j \{Y_j - \Psi^\top(S_j) \beta_{1,A_j}^* + \gamma \Psi^\top(S_{j+1}) \beta_{1,1}^*\} \end{array} \right\} \right],
\end{aligned}
$$

where the last equality is due to the definition of $\boldsymbol{\Sigma}_j$. Let

$$r_{a,j} = \Psi^\top(S_j) \beta_{a,A_t}^* - \gamma \Psi^\top(S_{j+1}) \beta_{a,0}^* - Q(a; A_j, S_j) + \gamma Q(a; a, S_{j+1}).$$

It follows that

$$\widehat{\boldsymbol{\beta}}(t) - \boldsymbol{\beta}^* = \widehat{\boldsymbol{\Sigma}}^{-1}(t) \left\{ \frac{1}{t} \sum_{j=0}^{t-1} \left( \begin{array}{c} \xi_j \varepsilon_{j,0} \\ \xi_j \varepsilon_{j,1} \end{array} \right) \right\} - \widehat{\boldsymbol{\Sigma}}^{-1}(t) \left\{ \frac{1}{t} \sum_{j=0}^{t-1} \left( \begin{array}{c} \xi_j r_{j,0} \\ \xi_j r_{j,1} \end{array} \right) \right\},$$

and hence

$$
\widehat{\boldsymbol{\beta}}(t) - \boldsymbol{\beta}^* = \boldsymbol{\Sigma}^{-1}(t) \left\{ \frac{1}{t} \sum_{j=0}^{t-1} \left( \begin{array}{c} \xi_j \varepsilon_{j,0} \\ \xi_j \varepsilon_{j,1} \end{array} \right) \right\} + \underbrace{\{\widehat{\boldsymbol{\Sigma}}^{-1}(t) - \boldsymbol{\Sigma}^{-1}(t)\} \left\{ \frac{1}{t} \sum_{j=0}^{t-1} \left( \begin{array}{c} \xi_j \varepsilon_{j,0} \\ \xi_j \varepsilon_{j,1} \end{array} \right) \right\}}_{\zeta_2(t)}
$$

$$
- \underbrace{\widehat{\boldsymbol{\Sigma}}^{-1}(t) \left\{ \frac{1}{t} \sum_{j=0}^{t-1} \left( \begin{array}{c} \xi_j r_{j,0} \\ \xi_j r_{j,1} \end{array} \right) \right\}}_{\zeta_3(t)}.
$$

We first consider $\zeta_3(t)$. It can be upper bounded by

$$
\|\widehat{\boldsymbol{\Sigma}}^{-1}(t)\|_2 \left\| \frac{1}{t} \sum_{j=0}^{t-1} \left( \begin{array}{c} \xi_j \varepsilon_{j,0} \\ \xi_j \varepsilon_{j,1} \end{array} \right) \right\|_2 = \|\widehat{\boldsymbol{\Sigma}}^{-1}(t)\|_2 \max_{a \in \{0,1\}} \sup_{\|\boldsymbol{a}\|_2 = 1} \left| \boldsymbol{a}^\top \left( \frac{1}{t} \sum_{j=0}^{t-1} \xi_j r_{a,j} \right) \right|
$$

$$
\le \|\widehat{\boldsymbol{\Sigma}}^{-1}(t)\|_2 \max_{a \in \{0,1\}} \sup_{\|\boldsymbol{a}\|_2 = 1} \sqrt{\frac{1}{t} \sum_{j=0}^{t-1} (\boldsymbol{a}^\top \xi_j)^2 r_{a,j}^2} \le \max_{a,j} |r_{a,j}| \|\widehat{\boldsymbol{\Sigma}}^{-1}(t)\|_2 \sqrt{\lambda_{\max} \left( \frac{1}{t} \sum_{j=0}^{t-1} \xi_j \xi_j^\top \right)},
$$

where the second follows from Cauchy-Schwarz inequality. Under Condition C2(i), we have for any $j \leq t \leq T_k$, $\max_j |r_{a,j}| = o(t^{-1/2})$. Suppose for now, we have shown

$$\|\widehat{\boldsymbol{\Sigma}}^{-1}(t)\|_2 = O_p(1) \quad \text{and} \quad \lambda_{\max}\left(\frac{1}{t}\sum_{j=0}^{t-1}\xi_j\xi_j^\top\right) = O_p(1). \tag{17}$$

It follows that

$$\zeta_3(t) = o_p(t^{-1/2}). \tag{18}$$

To bound $\zeta_2(t)$, notice that for any $a \in \{0,1\}$,

$$\mathrm{E}\left\|\frac{1}{t}\sum_{j=0}^{t-1}\xi_j\varepsilon_{j,a}\right\|_2^2 = \frac{1}{t^2}\sum_{j=0}^{t-1}\mathrm{E}\xi_j^\top\xi_j\varepsilon_{j,a}^2 + \frac{1}{t^2}\sum_{j_1 \neq j_2}\mathrm{E}\xi_{j_1}^\top\xi_{j_2}\varepsilon_{j_1,a}\varepsilon_{j_2,a}.$$

Similar to Lemma 1, we can show for any $\varphi(\cdot)$ that is a function of $\bar{A}_t, \bar{S}_t, \bar{Y}_{t-1}$ that

$$\mathrm{E}\{Q(a';A_t,S_t) - Y_t - \gamma Q(a';a',S_{t+1})\}\varphi(\bar{S}_t,\bar{A}_t,\bar{Y}_{t-1}) = 0. \tag{19}$$

This implies that $\mathrm{E}\xi_{j_1}^\top\xi_{j_2}\varepsilon_{j_1,a}\varepsilon_{j_2,a} = 0$ for any $j_1 \neq j_2$. It follows that

$$\mathrm{E}\left\|\frac{1}{t}\sum_{j=0}^{t-1}\xi_j\varepsilon_{j,a}\right\|_2^2 = \frac{1}{t^2}\sum_{j=0}^{t-1}\mathrm{E}\xi_j^\top\xi_j\varepsilon_{j,a}^2 \leq q\lambda_{\max}\left(\frac{1}{t^2}\sum_{j=0}^{t-1}\mathrm{E}\xi_j\xi_j^\top\varepsilon_{j,a}^2\right).$$

Since all immediate rewards are uniformly bounded, so is the Q function. As a result, $|\varepsilon_{j,a}|$'s are uniformly bounded. Suppose for now, we have shown

$$\lambda_{\max}\left(\frac{1}{t}\sum_{j=0}^{t-1}\mathrm{E}\xi_j\xi_j^\top\right) = O(1). \tag{20}$$

It follows that $\mathrm{E}\|t^{-1}\sum_{j=0}^{t-1}\xi_j\varepsilon_{j,a}\|_2^2 = O(q)$ and hence

$$\frac{1}{t}\sum_{j=0}^{t-1}\left(\begin{array}{c}\xi_j\varepsilon_{j,0}\\\xi_j\varepsilon_{j,1}\end{array}\right) = O_p(t^{-1/2}\sqrt{q}).$$

Suppose

$$\|\widehat{\boldsymbol{\Sigma}}^{-1}(t) - \boldsymbol{\Sigma}^{-1}(t)\|_2 = o_p(q^{-1/2}). \tag{21}$$

It follows that

$$\|\zeta_2(t)\|_2 \leq \|\widehat{\boldsymbol{\Sigma}}^{-1}(t) - \boldsymbol{\Sigma}^{-1}(t)\|_2\left\|\frac{1}{t}\sum_{j=0}^{t-1}\left(\begin{array}{c}\xi_j\varepsilon_{j,0}\\\xi_j\varepsilon_{j,1}\end{array}\right)\right\|_2 = o_p(t^{-1/2}).$$

This together with equation 18 yields equation 16.

It remains to show equation 17, equation 20 and equation 21 hold. We summarize these results in Lemma 3.

**Lemma 3** *Under the given conditions, we have equation 17, equation 20 and equation 21 hold.*

**Part 2:** By definition, we have $\widehat{\tau}(t) - \tau_0 = \boldsymbol{U}^\top\{\widehat{\boldsymbol{\beta}}(t) - \boldsymbol{\beta}^*\} + \boldsymbol{U}^\top\boldsymbol{\beta}^* - \tau_0$. Define

$$\boldsymbol{\Omega}(t) = \mathrm{E}\left\{\frac{1}{t}\sum_{j=0}^{t-1}\left(\begin{array}{c}\xi_j\varepsilon_{j,0}\\\xi_j\varepsilon_{j,1}\end{array}\right)\left(\begin{array}{c}\xi_j\varepsilon_{j,0}\\\xi_j\varepsilon_{j,1}\end{array}\right)^\top\right\}$$

The asymptotic variance of $\sqrt{t}\{\widehat{\tau}(t) - \tau_0\}$ is given by $\sigma^2(t) = \boldsymbol{U}^\top \boldsymbol{\Sigma}^{-1}(t)\boldsymbol{\Omega}(t)\{\boldsymbol{\Sigma}^{-1}(t)\}^\top \boldsymbol{U}$. We begin by providing a lower bound for $\sigma^2(t)$. Notice that

$$\sigma^2(t) \geq \lambda_{\min}\{\boldsymbol{\Omega}(t)\}\|\boldsymbol{U}^\top \boldsymbol{\Sigma}^{-1}(t)\|_2^2 \geq \lambda_{\min}\{\boldsymbol{\Omega}(t)\}\lambda_{\min}[\boldsymbol{\Sigma}^{-1}(t)\{\boldsymbol{\Sigma}^{-1}(t)\}^{-1}]\|\boldsymbol{U}\|_2^2. \tag{22}$$

Under C1(iii), we have $\liminf_q \|\boldsymbol{U}\|_2^2 > 0$.

In addition, notice that $\boldsymbol{\Sigma}^{-1}(t)\{\boldsymbol{\Sigma}^{-1}(t)\}^{-1}$ is positive semi-definite. It follows that $\lambda_{\min}[\boldsymbol{\Sigma}^{-1}(t)\{\boldsymbol{\Sigma}^{-1}(t)\}^{-1}] = 1/\lambda_{\max}[\boldsymbol{\Sigma}(t)\{\boldsymbol{\Sigma}(t)\}]$. Using similar arguments in showing $\|\boldsymbol{\Sigma}_{2,2}^{(0)*}(0)\|_2 = O(1)$ in the proof of Lemma 3, we can show $\sup_{t \geq 1}\|\boldsymbol{\Sigma}(t)\|_2 = O(1)$ and hence $\sup_{t \geq 1}\lambda_{\max}[\boldsymbol{\Sigma}(t)\{\boldsymbol{\Sigma}(t)\}] = O(1)$. This further yields

$$\inf_{t \geq 1} \lambda_{\min}[\boldsymbol{\Sigma}^{-1}(t)\{\boldsymbol{\Sigma}^{-1}(t)\}^{-1}] > 0. \tag{23}$$

Suppose $\boldsymbol{\Omega}(t)$ satisfies

$$\liminf_t \lambda_{\min}\{\boldsymbol{\Omega}(t)\} > 0. \tag{24}$$

It follows that $\sigma^2(t)$ is bounded away from zero, for sufficiently large $t$. Under Condition C2(i), we have $\boldsymbol{U}^\top \boldsymbol{\beta}^* - \tau_0 = o(T_k^{-1/2}) = o(t^{-1/2})$. It follows that

$$\frac{\sqrt{t}\{\widehat{\tau}(t) - \tau_0\}}{\sigma(t)} = \frac{\sqrt{t}\boldsymbol{U}^\top\{\widehat{\boldsymbol{\beta}}(t) - \boldsymbol{\beta}^*\}}{\sigma(t)} + \frac{\sqrt{t}(\boldsymbol{U}^\top\boldsymbol{\beta}^* - \tau_0)}{\sigma(t)} = \frac{\sqrt{t}\boldsymbol{U}^\top\{\widehat{\boldsymbol{\beta}}(t) - \boldsymbol{\beta}^*\}}{\sigma(t)} + o(1).$$

Moreover, it follows from equation 22, equation 23 and equation 24 that $\sigma(t)/\|\boldsymbol{U}\|_2$ is uniformly bounded away from zero, for sufficiently large $t$. Combining this together with equation 16 yields

$$\frac{\sqrt{t}\{\widehat{\tau}(t) - \tau_0\}}{\sigma(t)} = \frac{\sqrt{t}\boldsymbol{U}^\top\zeta_1(t)}{\sigma(t)} + \frac{\sqrt{t}\boldsymbol{U}^\top R_t}{\sigma(t)},$$

where the remainder term satisfies $\|R_t\|_2 = o_p(t^{-1/2})$. It follows that the second term on the right-hand-side (RHS) of the above expression is bounded from above by $\sqrt{t}\|R_t\|_2\|\boldsymbol{U}\|_2/\sigma(t) = o_p(1)$ and hence

$$\frac{\sqrt{t}\{\widehat{\tau}(t) - \tau_0\}}{\sigma(t)} = \frac{\sqrt{t}\boldsymbol{U}^\top\zeta_1(t)}{\sigma(t)} + o_p(1). \tag{25}$$

Similar to the proof of Lemma 1, we can show for any $j \geq 0$, $a \in \{0, 1\}$,

$$\mathrm{E}(\xi_j\varepsilon_{j,a}|\{S_i, A_i, Y_i\}_{i<j}) = 0.$$

By the definition of $\zeta_1(t)$, $\sqrt{t}\boldsymbol{U}^\top\zeta_1(t)/\sigma(t)$ forms a martingle with respect to the filtration $\sigma(\{S_j, A_j, Y_j\}_{j<t})$, i.e. the $\sigma$-algebra generated by $\{S_j, A_j, Y_j\}_{j<t}$. By the martingale central limit theorem, we can show $\sqrt{t}\boldsymbol{U}^\top\zeta_1(t)/\sigma(t) \xrightarrow{d} N(0, 1)$ (see Lemma 4 for details).

To complete the proof of Part 2, we need to show equation 24 holds and that $\widehat{\sigma}(t)/\sigma(t) \xrightarrow{P} 1$. The assertion $\widehat{\sigma}(t)/\sigma(t) \xrightarrow{P} 1$ can be similarly proven using arguments from Step 3 of the proof of Theorem 1, Shi et al. (2020a). We show the asymptotic normality of $\sqrt{t}\boldsymbol{U}^\top\zeta_1(t)/\sigma(t)$ and that equation 24 holds in the following lemma.

**Lemma 4** *Under the given conditions, we have equation 24 holds and that $\sqrt{t}\boldsymbol{U}^\top\zeta_1(t)/\sigma(t) \xrightarrow{d} N(0, 1)$.*

**Part 3:** Results in Part 2 yield that $\sqrt{T_k}\{\widehat{\tau}(T_k) - \tau_0\}/\sigma(T_k) \xrightarrow{d} N(0, 1)$ for each $1 \leq k \leq K$. In addition, for any $K$-dimensional vector $\boldsymbol{a} = (a_1, \cdots, a_K)^\top$, it follows from equation 25 that

$$\sum_{k=1}^K \frac{a_k\sqrt{T_k}\{\widehat{\tau}(T_k) - \tau_0\}}{\sigma(T_k)} = \sum_{k=1}^K \frac{a_k\sqrt{T_k}\boldsymbol{U}^\top\zeta_1(T_k)}{\sigma(T_k)} + o_p(1).$$

The leading term on the RHS can be rewritten as a weighted sum of $\{\xi_j\varepsilon_{j,0}, \xi_j\varepsilon_{j,1}\}_{0 \leq j < t}$. Similar to the proof in Part 2, we can show it forms a martingale with respect to the filtration

$\sigma(\{S_j, A_j, Y_j\}_{j<t})$. We now derive its asymptotic normality for any $\boldsymbol{a}$, using the martingale central limit theorem for triangular arrays.

By Corollary 2 of McLeish (1974), we need to verify the following two conditions:

(a) $\max_{0 \leq j < t} |\sum_{k=1}^{K} a_k T_k^{-1/2} \boldsymbol{U}^\top \boldsymbol{\Sigma}^{-1}(T_k)(\xi_j^\top \varepsilon_{j,0}, \xi_j^\top \varepsilon_{j,1})^\top \{\sigma(T_k)\}^{-1} \mathbb{I}(j < T_k)| \xrightarrow{P} 0$;

(b) $\sum_{j=0}^{T-1} |\sum_{k=1}^{K} a_k T_k^{-1/2} \boldsymbol{U}^\top \boldsymbol{\Sigma}^{-1}(T_k)(\xi_j^\top \varepsilon_{j,0}, \xi_j^\top \varepsilon_{j,1})^\top \{\sigma(T_k)\}^{-1} \mathbb{I}(j < T_k)|^2$ converges to some constant in probability.

Since $K$ is fixed, to verify (a), it suffices to show

$$\max_{1 \leq j < t, 1 \leq k \leq K} T_k^{-1/2} |\boldsymbol{U}^\top \boldsymbol{\Sigma}^{-1}(T_k)(\xi_j^\top \varepsilon_{j,0}, \xi_j^\top \varepsilon_{j,1})^\top \{\sigma(T_k)\}^{-1}| \xrightarrow{P} 0.$$

In Lemma 3, we have shown $\|\boldsymbol{\Sigma}^{-1}(t)\| = O(1)$. In Part 1 and Part 2 of the proof, we have shown $|\varepsilon_{j,a}|$'s are uniformly bounded and that $\sigma(t)/\|\boldsymbol{U}\|_2$ is bounded away from zero. Therefore, it suffices to show $T_1^{-1/2} \max_{0 \leq j < t} \|\xi_j\|_2 \xrightarrow{P} 0$. Under Condition C2(ii), we have $\sup_s \|\Psi(s)\|_2 = O(q^{1/2})$ and hence $\max_{0 \leq j < t} \|\xi_j\|_2 = O(q^{1/2})$. The assertion thus follows by noting that $T_1 = c_1 T$ and $q = o(T)$.

Using similar arguments in Step 3 of the proof of Shi et al. (2020a), we can show

$$\left\| \frac{1}{t} \sum_{j=0}^{t-1} (\xi_j^\top \varepsilon_{j,0}, \xi_j^\top \varepsilon_{j,1})^\top (\xi_j^\top \varepsilon_{j,0}, \xi_j^\top \varepsilon_{j,1}) - \boldsymbol{\Omega}(t) \right\|_2 \xrightarrow{P} 0, \tag{26}$$

as $t \to \infty$. This together with the facts $\|\boldsymbol{\Sigma}^{-1}(t)\| = O(1)$ and $\sigma(t)/\|\boldsymbol{U}\|_2$ is bounded away from zero implies that

$$\left| \frac{a_{k_1} a_{k_2}}{\sqrt{T_{k_1} T_{k_2}} \sigma^2(T_{k_1} \wedge T_{k_2})} \sum_{j=0}^{T_{k_1} \wedge T_{k_2}} \boldsymbol{U}^\top \boldsymbol{\Sigma}^{-1}(T_{k_1})(\xi_j^\top \varepsilon_{j,0}, \xi_j^\top \varepsilon_{j,1})^\top (\xi_j^\top \varepsilon_{j,0}, \xi_j^\top \varepsilon_{j,1}) \{\boldsymbol{\Sigma}^{-1}(T_{k_2})\}^\top \boldsymbol{U} \right.$$

$$\left. - \frac{a_{k_1} a_{k_2} (T_{k_1} \wedge T_{k_2})}{\sqrt{T_{k_1} T_{k_2}} \sigma^2(T_{k_1} \wedge T_{k_2})} \boldsymbol{U}^\top \boldsymbol{\Sigma}^{-1}(T_{k_1}) \boldsymbol{\Omega}(T_{k_1} \wedge T_{k_2}) \{\boldsymbol{\Sigma}^{-1}(T_{k_2})\}^\top \boldsymbol{U} \right\|_2$$

$$\leq \frac{a_{k_1} a_{k_2}}{\sigma^2(T_{k_1} \wedge T_{k_2})} \|\boldsymbol{U}\|_2^2 \max_k \|\boldsymbol{\Sigma}^{-1}(T_k)\|_2^2 \left\| \frac{1}{T_{k_1} \wedge T_{k_2}} \sum_{j=0}^{T_{k_1} \wedge T_{k_2} - 1} (\xi_j^\top \varepsilon_{j,0}, \xi_j^\top \varepsilon_{j,1})^\top (\xi_j^\top \varepsilon_{j,0}, \xi_j^\top \varepsilon_{j,1}) - \boldsymbol{\Omega}(t) \right|$$

$$\xrightarrow{P} 0,$$

where $a \wedge b = \min(a, b)$. It follows that

$$\left| \sum_{j=0}^{T-1} \left| \sum_{k=1}^{K} a_k T_k^{-1/2} \boldsymbol{U}^\top \boldsymbol{\Sigma}^{-1}(T_k)(\xi_j^\top \varepsilon_{j,0}, \xi_j^\top \varepsilon_{j,1})^\top \{\sigma(T_k)\}^{-1} \mathbb{I}(j < T_k) \right|^2 \right. \tag{27}$$

$$\left. - \sum_{k_1 \neq k_2} \frac{a_{k_1} a_{k_2} (T_{k_1} \wedge T_{k_2})}{\sqrt{T_{k_1} T_{k_2}} \sigma^2(T_{k_1} \wedge T_{k_2})} \boldsymbol{U}^\top \boldsymbol{\Sigma}^{-1}(T_{k_1}) \boldsymbol{\Omega}(T_{k_1} \wedge T_{k_2}) \{\boldsymbol{\Sigma}^{-1}(T_{k_2})\}^\top \boldsymbol{U} \right| = o_p(1).$$

In the proofs of Lemmas 3 and 4, we show that $\|\boldsymbol{\Sigma}^{-1}(t) - (\boldsymbol{\Sigma}^{(0)*})^{-1}\|_2 = O(t^{-1/2})$ and $\|\boldsymbol{\Omega}(t) - \boldsymbol{\Omega}^{(0)*}\|_2 = O(t^{-1/2})$ for some matrices $\boldsymbol{\Sigma}^{(0)*}$ and $\boldsymbol{\Omega}^{(0)*}$ that are invariant to $t$. Definitions of these two matrices can be found in Sections F.5 and F.6. Under C2(ii) and the condition that $q = o(\sqrt{T}/\log T)$, we can show $\|\boldsymbol{U}\|_2 = O(q^{1/2})$ and hence $\sigma^2(t) \xrightarrow{P} (\sigma^{(0)*})^2$ where $(\sigma^{(0)*})^2 = \boldsymbol{U}^\top (\boldsymbol{\Sigma}^{(0)*})^{-1} \boldsymbol{\Omega}^{(0)*} \{(\boldsymbol{\Sigma}^{(0)*})^{-1}\}^\top \boldsymbol{U}$. Similar to equation 27, we have

$$\sum_{k_1 \neq k_2} \frac{a_{k_1} a_{k_2} (T_{k_1} \wedge T_{k_2})}{\sqrt{T_{k_1} T_{k_2}} \sigma^2(T_{k_1} \wedge T_{k_2})} \boldsymbol{U}^\top \boldsymbol{\Sigma}^{-1}(T_{k_1}) \boldsymbol{\Omega}(T_{k_1} \wedge T_{k_2}) \{\boldsymbol{\Sigma}^{-1}(T_{k_2})\}^\top \boldsymbol{U} \tag{28}$$

$$\xrightarrow{P} \sum_{k_1 \neq k_2} \frac{a_{k_1} a_{k_2} (T_{k_1} \wedge T_{k_2})}{\sqrt{T_{k_1} T_{k_2}} (\sigma^{(0)*})^2} \boldsymbol{U}^\top \{\boldsymbol{\Sigma}^{(0)*}\}^{-1} \boldsymbol{\Omega}^{(0)*} \{(\boldsymbol{\Sigma}^{(0)*})^{-1}\}^\top \boldsymbol{U} \to \sum_{k_1 \neq k_2} \frac{a_{k_1} a_{k_2} (c_{k_1} \wedge c_{k_2})}{\sqrt{c_{k_1} c_{k_2}}},$$

where $c_k$'s are defined in Section 4.4. This together with equation 27 yields that

$$\sum_{j=0}^{T-1} \left| \sum_{k=1}^{K} a_k T_k^{-1/2} \boldsymbol{U}^\top \boldsymbol{\Sigma}^{-1}(T_k)(\xi_j^\top \varepsilon_{j,0}, \xi_j^\top \varepsilon_{j,1})^\top \{\sigma(T_k)\}^{-1} \mathbb{I}(j < T_k) \right|^2 \xrightarrow{P} \frac{a_{k_1} a_{k_2}(c_{k_1} \wedge c_{k_2})}{\sqrt{c_{k_1} c_{k_2}}}.$$

Conditions (a) and (b) are thus verified. By Lemma 4, we can show

$$\sum_{k=1}^{K} \frac{a_k \sqrt{T_k} \{\widehat{\tau}(T_k) - \tau_0\}}{\sigma(T_k)} = \sum_{k=1}^{K} \frac{a_k \sqrt{T_k} \{\widehat{\tau}(T_k) - \tau_0\}}{\widehat{\sigma}(T_k)} + o_p(1),$$

for any $(a_1, \cdots, a_K)$. This yields the joint asymptotic normality of our test statistics.

By equation 28, its covariance matrix is given by $\boldsymbol{\Xi}_0$ whose $(k_1, k_2)$-th entry is equal to $(c_{k_1} c_{k_2})^{-1/2} c_{k_1} \wedge c_{k_2}$. Using similar arguments in proving equation 27, equation 28 and Step 3 of the proof in Theorem 1, Shi et al. (2020a), we can show $\widehat{\boldsymbol{\Xi}}$ is a consistent estimator for $\boldsymbol{\Xi}_0$. This completes the proof of Theorem 3 under D1.

### F.2.2 PROOF UNDER D2

The proof is very similar to that under D1. Suppose we can show equation 17, equation 20, equation 21 and equation 24 hold. Then similar to the proof under D1, we have

$$\frac{\sqrt{t}\{\widehat{\tau}(t) - \tau_0\}}{\sigma(t)} = \frac{\sqrt{t} \boldsymbol{U}^\top \zeta_1(t)}{\sigma(t)} + o_p(1).$$

The following lemma shows these assertions hold under D2 as well.

**Lemma 5** *Under the given conditions, we have equation 17, equation 20, equation 21 and equation 24 hold.*

It follows that for any $K$-dimensional vector $\boldsymbol{a} = (a_1, \cdots, a_K)^\top$,

$$\sum_{k=1}^{K} \frac{a_k \sqrt{T_k} \{\widehat{\tau}(T_k) - \tau_0\}}{\sigma(T_k)} = \sum_{k=1}^{K} \frac{a_k \sqrt{T_k} \boldsymbol{U}^\top \zeta_1(T_k)}{\sigma(T_k)} + o_p(1).$$

In the proof of Lemma 5, we show $\|\{\boldsymbol{\Sigma}(t)\}^{-1}\|_2 = O(1)$, $\|\boldsymbol{\Sigma}(t) - \boldsymbol{\Sigma}^*\|_2 = O(t^{-1/2})$ for some time-invariant matrix $\boldsymbol{\Sigma}^*$ that satisfies $\|(\boldsymbol{\Sigma}^*)^{-1}\|_2 = O(1)$. It follows that

$$\|\{\boldsymbol{\Sigma}(t)\}^{-1} - (\boldsymbol{\Sigma}^*)^{-1}\|_2 = \|\{\boldsymbol{\Sigma}(t)\}^{-1}(\boldsymbol{\Sigma}(t) - \boldsymbol{\Sigma}^*)(\boldsymbol{\Sigma}^*)^{-1}\|_2 \leq$$
$$\|\{\boldsymbol{\Sigma}(t)\}^{-1}\|_2 \|\boldsymbol{\Sigma}(t) - \boldsymbol{\Sigma}^*\|_2 \|(\boldsymbol{\Sigma}^*)^{-1}\|_2 = O(t^{-1/2}).$$

Similarly, we can show $\|\boldsymbol{\Omega}(t) - \boldsymbol{\Omega}^*\|_2 = O(t^{-1/2})$ for some matrix $\boldsymbol{\Omega}^*$.

In addition, using similar arguments in the proof of Lemma 5, we can show equation 26 holds under D2 as well. Now, the joint asymptotic normality of our test statistics follow using arguments from Part 3 of the proof under D1. Similarly, we can show $\widehat{\boldsymbol{\Xi}}$ is consistent. This completes the proof under D2.

### F.2.3 PROOF UNDER D3

The proof under D1 indicates that equation 17, equation 20, equation 21 and equation 24 hold with $t = T_1$. It follows that

$$\frac{\sqrt{T_1}\{\widehat{\tau}(T_1) - \tau_0\}}{\sigma(T_1)} = \frac{\sqrt{T_1} \boldsymbol{U}^\top \zeta_1(T_1)}{\sigma(T_1)} + o_p(1). \tag{29}$$

The rest of the proof is divided into two parts. In the first part, we show for $k = 2, \cdots, K$,

$$\frac{\sqrt{T_k}\{\widehat{\tau}(T_k) - \tau_0\}}{\sigma^*(T_k)} = \frac{\sqrt{T_k} \boldsymbol{U}^\top \zeta_1^*(T_k)}{\sigma^*(T_k)} + o_p(1), \tag{30}$$

for some $\zeta_1^*(T_k)$ and $\sigma^*(T_k)$ defined below. In the second part, we show the assertion in Theorem 3 holds under D3.

**Part 1:** For any $1 \le k \le K$, consider the matrices

$$\mathbf{\Sigma}^{(k)} = \frac{1}{T_k - T_{k-1}} \sum_{j=T_{k-1}}^{T_k-1} \mathrm{E}[\mathbf{\Sigma}_j | \{(S_t, A_t, Y_t)\}_{0 \le t < T_{k-1}}] \quad \text{and} \quad \widehat{\mathbf{\Sigma}}^{(k)} = \frac{1}{T_k - T_{k-1}} \sum_{j=T_{k-1}}^{T_k-1} \mathbf{\Sigma}_j.$$

We show in Lemma 6 below that for $k = 2, \cdots, K$,

$$\|\mathbf{\Sigma}^{(k)} - \widehat{\mathbf{\Sigma}}^{(k)}\|_2 = o_p(q^{-1/2}), \tag{31}$$

and

$$\|\{\overline{\mathbf{\Sigma}}^{(k)}\}^{-1}\|_2 = O_p(1). \tag{32}$$

where $\overline{\mathbf{\Sigma}}^{(k)} = T_k^{-1} \sum_{i=1}^{k} (T_i - T_{i-1}) \mathbf{\Sigma}^{(i)}$.

**Lemma 6** *Under the given conditions, we have equation 31 and equation 32 hold.*

Notice that $(T_i - T_{i-1})/T_k \to (c_i - c_{i-1})/c_k$ and $\|\{c_k^{-1} \sum_{i=1}^{k} (c_i - c_{i-1}) \mathbf{\Sigma}^{(i)}\}^{-1}\|_2 = O_p(1)$ where $c_0 = 0$. It follows from equation 31 that $\|c_k^{-1} \sum_{i=1}^{k} (c_i - c_{i-1})(\widehat{\mathbf{\Sigma}}^{(i)} - \mathbf{\Sigma}^{(i)})\|_2 = o_p(q^{-1/2})$ and hence

$$\left\| \frac{1}{T_k} \sum_{i=1}^{k} (T_i - T_{i-1})(\widehat{\mathbf{\Sigma}}^{(i)} - \mathbf{\Sigma}^{(i)}) \right\|_2 = o_p(q^{-1/2}), \quad \forall k = 2, \cdots, K.$$

Similar to the proof under D1, we can show

$$\left\| \left\{ \frac{1}{T_k} \sum_{i=1}^{k} (T_i - T_{i-1}) \widehat{\mathbf{\Sigma}}^{(i)} \right\}^{-1} \right\|_2 = O_p(1) \quad \text{and} \quad \left\| \left\{ \frac{1}{T_k} \sum_{i=1}^{k} (T_i - T_{i-1}) \widehat{\mathbf{\Sigma}}^{(i)} \right\}^{-1} - \{\overline{\mathbf{\Sigma}}^{(k)}\}^{-1} \right\|_2 = o_p(q^{-1/2}),$$

for $k = 2, \cdots, K$. Thus, equation 21 and the first assertion in equation 17 hold with $t = T_2, T_3, \cdots, T_K$ under D3.

In addition, similar to Lemma 6, we can show

$$\lambda_{\max} \left[ \frac{1}{T_k - T_{k-1}} \sum_{j=T_{k-1}}^{T_k-1} \mathrm{E}[\xi_j \xi_j^\top | \{(S_t, A_t, Y_t)\}_{0 \le t < T_{k-1}}] \right] = O_p(1), \tag{33}$$

and

$$\left\| \frac{1}{T_k - T_{k-1}} \sum_{j=T_{k-1}}^{T_k-1} \left( \xi_j \xi_j^\top - \mathrm{E}[\xi_j \xi_j^\top | \{(S_t, A_t, Y_t)\}_{0 \le t < T_{k-1}}] \right) \right\|_2 = o_p(q^{-1/2}),$$

for $k = 2, \cdots, K$. This yields $\|(T_k - T_{k-1})^{-1} \sum_{j=T_{k-1}}^{T_k-1} \xi_j \xi_j^\top\|_2 = O_p(1)$. As a result, the second assertion in equation 17 holds with $t = T_2, T_3, \cdots, T_K$.

Moreover, using similar arguments in showing $t^{-1} \sum_{j=0}^{t-1} (\xi_j^\top \varepsilon_{j,0}, \xi_j^\top \varepsilon_{j,1})^\top = O_p(t^{-1/2}\sqrt{q})$ under D1, we have by equation 33 that $(T_k - T_{k-1})^{-1} \sum_{j=T_{k-1}}^{T_k-1} (\xi_j^\top \varepsilon_{j,0}, \xi_j^\top \varepsilon_{j,1})^\top = O_p\{(T_k - T_{k-1})^{-1/2}\sqrt{q}\}$, for $k = 1, \cdots, K$. Under the given conditions on $\{T_k\}_k$, we obtain $t^{-1} \sum_{j=0}^{t-1} (\xi_j^\top \varepsilon_{j,0}, \xi_j^\top \varepsilon_{j,1})^\top = O_p(t^{-1/2}\sqrt{q})$ for $t = T_2, T_3, \cdots, T_K$.

Based on these results, using similar arguments in Part 1 of the proof under D1, we can show

$$\sqrt{T_k} \{\widehat{\boldsymbol{\beta}}(T_k) - \boldsymbol{\beta}^*\} = \sqrt{T_k} \zeta_1^*(T_k) + o_p(1), \quad \forall k \in \{2, \cdots, K\}, \tag{34}$$

where

$$\zeta_1^*(T_k) = \frac{1}{T_k} \sum_{j=1}^{T_k} (\overline{\mathbf{\Sigma}}^{(k)})^{-1} \begin{pmatrix} \xi_j \varepsilon_{j,0} \\ \xi_j \varepsilon_{j,1} \end{pmatrix}.$$

For $1 \leq k \leq K$, define

$$\boldsymbol{\Omega}^{(k)} = \frac{1}{T_k - T_{k-1}} \sum_{j=T_{k-1}}^{T_k-1} \mathrm{E}[(\xi_j^\top \varepsilon_{j,0}, \xi_j^\top \varepsilon_{j,1})^\top (\xi_j^\top \varepsilon_{j,0}, \xi_j^\top \varepsilon_{j,1})|\{(S_t, A_t, Y_t)\}_{0 \leq t < T_{k-1}}],$$

and $\overline{\boldsymbol{\Omega}}^{(k)} = T_k^{-1} \sum_{i=1}^k (T_i - T_{i-1})\boldsymbol{\Sigma}^{(i)}$. For any $2 \leq k \leq K$, we have $\lambda_{\min}(\overline{\boldsymbol{\Omega}}^{(k)}) \geq \lambda_{\min}(T_k^{-1}T_1\boldsymbol{\Omega}^{(1)})$. Since $T_k^{-1}T_1 \to c_k^{-1}c_1 > 0$ and $\lambda_{\min}(\boldsymbol{\Omega}^{(1)}) = \lambda_{\min}(\boldsymbol{\Omega}(T_1))$ is bounded away from zero, $\lambda_{\min}(\overline{\boldsymbol{\Omega}}^{(k)})$ is bounded away from zero for $k = 2, \cdots, K$ as well. Define

$$\{\sigma^*(T_k)\}^2 = \boldsymbol{U}^\top (\overline{\boldsymbol{\Sigma}}^{(k)})^{-1} \overline{\boldsymbol{\Omega}}^{(k)} \{(\overline{\boldsymbol{\Sigma}}^{(k)})^{-1}\}^\top \boldsymbol{U}.$$

It can be shown that $\sigma^*(T_k)/\|\boldsymbol{U}\|_2$ is bounded away from zero, for $k = 2, \cdots, K$. Using similar arguments in Part 2 of the proof under D1, we can show equation 30 holds. This completes the proof for Part 1.

**Part 2:** Let $\sigma^*(T_1) = \sigma(T_1)$. By equation 29 and equation 30, we have for any $K$-dimensional vector $\boldsymbol{a} = (a_1, \cdots, a_K)^\top$ that

$$\sum_{k=1}^K \frac{a_k \sqrt{T_k}\{\widehat{\tau}(T_k) - \tau_0\}}{\sigma^*(T_k)} = \sum_{k=1}^K \frac{a_k \sqrt{T_k}\boldsymbol{U}^\top \zeta_1(T_k)}{\sigma^*(T_k)} + o_p(1). \tag{35}$$

In the following, we show the leading term on the RHS of equation 35 is asymptotically normal. Similar to the proof under D1, it suffices to verify the following conditions:

(a) $\max_{0 \leq j < T} |\sum_{k=1}^K a_k T_k^{-1/2} \boldsymbol{U}^\top (\overline{\boldsymbol{\Sigma}}^{(k)})^{-1}(\xi_j^\top \varepsilon_{j,0}, \xi_j^\top \varepsilon_{j,1})^\top \mathbb{I}(j < T_k)| \xrightarrow{P} 0$;

(b) $\sum_{j=0}^{T-1} |\sum_{k=1}^K a_k T_k^{-1/2} \boldsymbol{U}^\top (\overline{\boldsymbol{\Sigma}}^{(k)})^{-1}(\xi_j^\top \varepsilon_{j,0}, \xi_j^\top \varepsilon_{j,1})^\top \{\sigma^*(T_k)\}^{-1} \mathbb{I}(j < T_k)|^2$ converges to some constant in probability.

Condition (a) can be proven in a similar manner as in Part 3 of the proof under D1. Notice that for $k = 2, \cdots, K$, $\overline{\boldsymbol{\Sigma}}^{(k)}$, $\overline{\boldsymbol{\Omega}}^{(k)}$ and $\sigma^*(T_k)$ are random variables and depend on the observed data history. In the proof of Lemma 6, we show $\|(\overline{\boldsymbol{\Sigma}}^{(k)})^{-1} - (\boldsymbol{\Sigma}^{**})^{-1}\|_2 = O_p(T^{-1/2})$ for some deterministic matrix $\boldsymbol{\Sigma}^*$ and all $k \in \{2, \cdots, K\}$. Similarly, we can show $\|\overline{\boldsymbol{\Omega}}^{(k)} - \boldsymbol{\Omega}^{**}\|_2 = O_p(T^{-1/2})$ and $\|\{\sigma^*(T_k)\}^2 - (\sigma^{**})^2\|_2 = O_p(T^{-1/2})$ for some $\boldsymbol{\Sigma}^*$, $\sigma^{**}$ and all $k \in \{2, \cdots, K\}$. Moreover, using similar arguments in the proof of Lemma 6, we can show

$$\left\| \frac{1}{T_k - T_{k-1}} \sum_{j=T_{k-1}}^{T_k-1} \begin{pmatrix} \xi_j \varepsilon_{j,0} \\ \xi_j \varepsilon_{j,1} \end{pmatrix} \begin{pmatrix} \xi_j \varepsilon_{j,0} \\ \xi_j \varepsilon_{j,1} \end{pmatrix}^\top - \boldsymbol{\Omega}^{(k)} \right\|_2 = o_p(q^{-1/2}), \quad \forall k = 2, \cdots, K.$$

This further implies that

$$\left\| \frac{1}{T_k} \sum_{j=0}^{T_k-1} \begin{pmatrix} \xi_j \varepsilon_{j,0} \\ \xi_j \varepsilon_{j,1} \end{pmatrix} \begin{pmatrix} \xi_j \varepsilon_{j,0} \\ \xi_j \varepsilon_{j,1} \end{pmatrix}^\top - \boldsymbol{\Omega}^{**} \right\|_2 = o_p(q^{-1/2}), \quad \forall k = 2, \cdots, K.$$

Based on these results, using similar arguments in Part 3 of the proof of Lemma 3, we obtain (b). The joint asymptotic normality of $\sqrt{T_1}\{\widehat{\tau}(T_1) - \tau_0\}/\sigma^*(T_1), \cdots, \sqrt{T_1}\{\widehat{\tau}(T_1) - \tau_0\}/\sigma^*(T_K)$ thus follows.

Consistency of $\widehat{\boldsymbol{\Xi}}$ can be similarly proven. We omit the details for brevity.

### F.3 PROOF OF THEOREM 1

As discussed in Section 4.3, $(Z_1^*, Z_2^*, \cdots, Z_K^*)^\top$ is jointly normal with mean zero and covariance matrix $\widehat{\boldsymbol{\Xi}}$, conditional on the observed data. By Theorem 1, we have $\widehat{\boldsymbol{\Xi}} \xrightarrow{P} \boldsymbol{\Xi}_0$ where $\boldsymbol{\Xi}_0$ is the asymptotic covariance matrix of $(Z_1, Z_2, \cdots, Z_K)^\top$. Let $\alpha^*(t) = \alpha(tT)$ for any $0 \leq t \leq 1$, we have $\alpha(T_k) \to \alpha^*(c_k)$ for any $1 \leq k \leq K$. Notice that $\{\widehat{b}_k\}_{1 \leq k \leq K}$ is a continuous function of $\widehat{\boldsymbol{\Xi}}$

and $\{\alpha(T_k)\}_{1 \le k \le K}$, it follows that $\widehat{b}_k \xrightarrow{P} b_{k,0}$ for $1 \le k \le K$, where $\{b_{k,0}\}_{1 \le k \le K}$ are recursively defined as follows:

$$\Pr\left\{ \max_{1 \le j < k}(Z_{j,0} - b_{j,0}) \le 0, Z_{k,0} > b_{k,0} \right\} = \alpha^*(c_k) - \alpha^*(c_{k-1}),$$

where $(Z_{1,0}, Z_{2,0}, \cdots, Z_{K,0})^\top$ is asymptotically normal with mean zero and covariance matrix $\mathbf{\Xi}_0$.

Theorem 3 implies that $(Z_1 - \sqrt{T_1}\tau_0/\widehat{\sigma}(T_1), Z_2 - \sqrt{T_2}\tau_0/\widehat{\sigma}(T_2), \cdots, Z_K - \sqrt{T_K}\tau_0/\widehat{\sigma}(T_K))^\top \xrightarrow{d} (Z_{1,0}, Z_{2,0}, \cdots, Z_{K,0})^\top$. It follows that

$$\Pr\left( \bigcup_{j=1}^{k}\{Z_j > \widehat{b}_j\} \right) \le \Pr\left( \bigcup_{j=1}^{k}\{Z_j - \sqrt{T_j}\tau_0/\widehat{\sigma}(T_j) > \widehat{b}_j\} \right) \to \Pr\left( \bigcup_{j=1}^{k}\{Z_{j,0} > b_{j,0}\} \right) = \alpha^*(c_k). \quad (36)$$

The proof is hence completed by noting that $\alpha(T_k) \to \alpha^*(c_k)$. When $\tau_0 = 0$, we have $\mathrm{E}Z_k = o(1)$. The rejection probability thus converges to the nominal level.

## F.4 Proof of Theorem 2

Suppose $\tau_0 = T^{-1/2}h$ for some $h > 0$. Based on the proof of Theorem 3, we can show $\widehat{\sigma}(T_k) \xrightarrow{P} \sigma_k^*$ for some $\sigma_k^* > 0$. It follows from equation 36 that

$$\Pr\left( \bigcup_{j=1}^{k}\{Z_j > \widehat{b}_j\} \right) = \Pr\left( \bigcup_{j=1}^{k}\{Z_j - \sqrt{T_j}\tau_0/\widehat{\sigma}(T_j) > \widehat{b}_j - h/\widehat{\sigma}(T_j)\} \right)$$

$$\to \Pr\left( \bigcup_{j=1}^{k}\{Z_{j,0} > b_{j,0} - h/\sigma_j^*\} \right) > \alpha^*(c_k).$$

The second assertion in Theorem 2 thus holds by noting that $\alpha(T_k) \to \alpha^*(c_k)$.

Let $h \to \infty$, we obtain

$$\Pr\left( \bigcup_{j=1}^{k}\{Z_j > \widehat{b}_j\} \right) = \Pr\left( \bigcup_{j=1}^{k}\{Z_{j,0} > b_{j,0} - h/\sigma_j^*\} \right) + o(1) \to 1.$$

The proof is hence completed.

## F.5 Proof of Lemma 3

Under the given conditions in C1(i), C1(ii) and C2(ii), equation 20 and the second assertion in equation 17 can be proven using similar arguments in the proof of Lemma E.2 and E.3 of Shi et al. (2020a). We omit the proof for brevity.

It remains to show equation 21 and the first assertion in equation 17 hold. Recall that $\mu$ is the density function of the stationary distribution $\Pi$ (see the remark below Condition C1). In addition, $\mu$ is uniformly bounded away from 0 and $\infty$ under C1(i). For $a' \in \{0, 1\}$, define the matrix

$$\mathbf{\Sigma}^{(0)*}(a') = \int_{s,s' \in \mathbb{S}} \sum_{a \in \{0,1\}} \xi(s,a)\{\xi(s,a) - \gamma\xi(s',a')\}\mu(s)\{(1-a)b^{(0)}(s) + a(1-b^{(0)}(s))\}p(s';a,s)dsds'.$$

Define

$$\mathbf{\Sigma}^{(0)*} = \begin{bmatrix} \mathbf{\Sigma}^{(0)*}(0) & \\ & \mathbf{\Sigma}^{(0)*}(1) \end{bmatrix}.$$

The matrix $\mathbf{\Sigma}^{(0)*}$ is the population limit of $\widehat{\mathbf{\Sigma}}(t)$ under D1. To prove the first assertion in equation 17, we first show

$$\|(\mathbf{\Sigma}^{(0)*})^{-1}\|_2 = O(1). \quad (37)$$

By definition, this is equivalent to show

$$\|\{\mathbf{\Sigma}^{(0)*}(a)\}^{-1}\|_2 = O(1),$$

for $a \in \{0, 1\}$. The matrix $\mathbf{\Sigma}^{(0)*}(0)$ can be written as

$$\mathbf{\Sigma}^{(0)*}(0) = \left[ \begin{array}{cc} \mathbf{\Sigma}_{1,1}^{(0)*}(0) & \\ \mathbf{\Sigma}_{2,1}^{(0)*}(0) & \mathbf{\Sigma}_{2,2}^{(0)*}(0) \end{array} \right],$$

where

$$\begin{aligned}
\mathbf{\Sigma}_{1,1}^{(0)*}(0) &= \int_{s,s' \in \mathbb{S}} \Psi(s)\{\Psi(s) - \gamma\Psi(s')\}^\top b^{(0)}(s)\mu(s)p(s'; 0, s)dsds', \\
\mathbf{\Sigma}_{2,1}^{(0)*}(0) &= -\gamma \int_{s,s' \in \mathbb{S}} \Psi(s)\Psi^\top(s')(1 - b^{(0)}(s))\mu(s)p(s'; 0, s)dsds', \\
\mathbf{\Sigma}_{2,2}^{(0)*}(0) &= \int_{s \in \mathbb{S}} \Psi(s)\Psi^\top(s)\mu(s)(1 - b^{(0)}(s))p(s'; 1, s)ds.
\end{aligned}$$

It follows that

$$\{\mathbf{\Sigma}^{(0)*}(0)\}^{-1} = \left[ \begin{array}{cc} \{\mathbf{\Sigma}_{1,1}^{(0)*}(0)\}^{-1} & \\ -\{\mathbf{\Sigma}_{2,2}^{(0)*}(0)\}^{-1}\mathbf{\Sigma}_{2,1}^{(0)*}(0)\{\mathbf{\Sigma}_{1,1}^{(0)*}(0)\}^{-1} & \{\mathbf{\Sigma}_{2,2}^{(0)*}(0)\}^{-1} \end{array} \right],$$

and hence

$$\begin{aligned}
\|\{\mathbf{\Sigma}^{(0)*}(0)\}^{-1}\|_2 &= \sup_{\|\boldsymbol{a}_1\|_2=1, \|\boldsymbol{a}_2\|_2=1} \left| \boldsymbol{a}_1^\top \left[ \begin{array}{cc} \{\mathbf{\Sigma}_{1,1}^{(0)*}(0)\}^{-1} & \\ -\{\mathbf{\Sigma}_{2,2}^{(0)*}(0)\}^{-1}\mathbf{\Sigma}_{2,1}^{(0)*}(0)\{\mathbf{\Sigma}_{1,1}^{(0)*}(0)\}^{-1} & \{\mathbf{\Sigma}_{2,2}^{(0)*}(0)\}^{-1} \end{array} \right] \boldsymbol{a}_2 \right| \\
&\leq \sup_{\|\boldsymbol{a}_3\|_2=1, \|\boldsymbol{a}_4\|_2=1} |\boldsymbol{a}_3^\top \{\mathbf{\Sigma}_{1,1}^{(0)*}(0)\}^{-1}\boldsymbol{a}_4| + \sup_{\|\boldsymbol{a}_3\|_2=1, \|\boldsymbol{a}_4\|_2=1} |\boldsymbol{a}_3^\top \{\mathbf{\Sigma}_{2,2}^{(0)*}(0)\}^{-1}\boldsymbol{a}_4| \\
&\quad + \sup_{\|\boldsymbol{a}_3\|_2=1, \|\boldsymbol{a}_4\|_2=1} |\boldsymbol{a}_3^\top \{\mathbf{\Sigma}_{2,2}^{(0)*}(0)\}^{-1}\mathbf{\Sigma}_{2,1}^{(0)*}(0)\{\mathbf{\Sigma}_{1,1}^{(0)*}(0)\}^{-1}\boldsymbol{a}_4| \\
&\leq \|\{\mathbf{\Sigma}_{1,1}^{(0)*}(0)\}^{-1}\|_2 + \|\{\mathbf{\Sigma}_{2,2}^{(0)*}(0)\}^{-1}\|_2 + \|\{\mathbf{\Sigma}_{2,2}^{(0)*}(0)\}^{-1}\|_2\|\mathbf{\Sigma}_{2,1}^{(0)*}(0)\|_2\|\{\mathbf{\Sigma}_{1,1}^{(0)*}(0)\}^{-1}\|_2.
\end{aligned}$$

Thus, to prove $\|\{\mathbf{\Sigma}^{(0)*}(0)\}^{-1}\|_2 = O(1)$, it suffices to show

$$\|\{\mathbf{\Sigma}_{1,1}^{(0)*}(0)\}^{-1}\|_2 = O(1), \tag{38}$$

$$\|\{\mathbf{\Sigma}_{2,2}^{(0)*}(0)\}^{-1}\|_2 = O(1), \tag{39}$$

$$\|\mathbf{\Sigma}_{2,1}^{(0)*}(0)\|_2 = O(1). \tag{40}$$

We first consider equation 38. Using similar arguments in Part 1 of the proof of Lemma E.2, Shi et al. (2020a), it suffices to show

$$\boldsymbol{a}^\top \mathbf{\Sigma}_{1,1}^{(0)*}(0)\boldsymbol{a} \geq \bar{c}_1\|\boldsymbol{a}\|_2^2, \quad \forall \boldsymbol{a},$$

for some $\bar{c}_1 > 0$. Under D1, $b^{(0)}$ is strictly positive. Since $\mu$ is strictly positive, it suffices to show

$$\boldsymbol{a}^\top \int_{s,s' \in \mathbb{S}} \Psi(s)\{\Psi(s) - \gamma\Psi(s')\}^\top dsds' \boldsymbol{a} \geq \bar{c}_2\|\boldsymbol{a}\|_2^2, \quad \forall \boldsymbol{a}, \tag{41}$$

for some $\bar{c}_2 > 0$. Notice that LHS of equation 41 is equal to

$$\lambda(\mathbb{S}) \int_{s \in \mathbb{S}} \{\boldsymbol{a}^\top \Psi(s)\}^2 ds - \gamma \int_{s,s' \in \mathbb{S}} \{\boldsymbol{a}^\top \Psi(s)\}\{\boldsymbol{a}^\top \Psi(s')\}dsds',$$

where $\lambda(\mathbb{S})$ is the Lebesgue measure of $\mathbb{S}$. Since $\mathbb{S}$ is compact, we have $\lambda(\mathbb{S}) < +\infty$. By Cauchy-Schwarz inequality, LHS of equation 41 is greater than or equal to

$$\lambda(\mathbb{S}) \int_{s \in \mathbb{S}} \{\boldsymbol{a}^\top \Psi(s)\}^2 ds - \lambda(\mathbb{S}) \int_{s \in \mathbb{S}} \frac{\gamma}{2}\{\boldsymbol{a}^\top \Psi(s)\}^2 ds - \lambda(\mathbb{S}) \int_{s \in \mathbb{S}} \frac{\gamma}{2}\{\boldsymbol{a}^\top \Psi(s')\}ds'$$

$$\geq (1 - \gamma)\lambda(\mathbb{S}) \int_{s \in \mathbb{S}} \{\boldsymbol{a}^\top \Psi(s)\}^2 ds.$$

This is directly implied by Condition C2(ii). The proof of equation 38 is hence completed. Similarly, we can prove equation 39. In addition, notice that

$$\|\mathbf{\Sigma}_{2,1}^{(0)*}(0)\|_2 \leq \sup_{\|\boldsymbol{a}_1\|_2=1, \|\boldsymbol{a}_2\|_2=1} \int_{s,s'\in\mathbb{S}} |\boldsymbol{a}_1^\top \Psi(s)||\boldsymbol{a}_2^\top \Psi(s')|\{1 - b^{(0)}(s)\mu(s)\}p(s';0,s)dsds'$$

$$\leq \sup_{\|\boldsymbol{a}_1\|_2=1, \|\boldsymbol{a}_2\|_2=1} \int_{s,s'\in\mathbb{S}} |\boldsymbol{a}_1^\top \Psi(s)||\boldsymbol{a}_2^\top \Psi(s')|\mu(s)p(s';0,s)dsds'.$$

Since the density function $\mu$ is uniformly bounded, we have

$$\|\mathbf{\Sigma}_{2,1}^{(0)*}(0)\|_2 \leq O(1) \sup_{\|\boldsymbol{a}_1\|_2=1, \|\boldsymbol{a}_2\|_2=1} \int_{s,s'\in\mathbb{S}} |\boldsymbol{a}_1^\top \Psi(s)||\boldsymbol{a}_2^\top \Psi(s')|dsds',$$

where $O(1)$ denotes the universal constant. By Cauchy-Schwarz inequality, we obtain

$$\|\mathbf{\Sigma}_{2,1}^{(0)*}(0)\|_2 \leq O(1)\lambda(\mathbb{S}) \sup_{\|\boldsymbol{a}\|_2=1} \int_{s\in\mathbb{S}} |\boldsymbol{a}^\top \Psi(s)|^2 ds \leq O(1)\lambda(\mathbb{S})\lambda_{\max}\left[\int_{s\in\mathbb{S}} \Psi(s)\Psi^\top(s)ds\right].$$

In view of C2(ii), we obtain equation 40.

To summarize, we have shown $\|\{\mathbf{\Sigma}^{(0)*}(0)\}^{-1}\|_2 = O(1)$. Similarly, we can prove $\|\{\mathbf{\Sigma}^{(0)*}(1)\}^{-1}\|_2 = O(1)$. Assertion equation 37 thus holds. Similar to Lemma E.5 of Shi et al. (2020a), we can show $\|\mathbf{\Sigma}(t) - \mathbf{\Sigma}^{(0)*}\|_2 = O(t^{-1/2})$. Using similar arguments in Part 1 of the proof of Lemma E.2, Shi et al. (2020a), this yields $\|\mathbf{\Sigma}^{-1}(t) - (\mathbf{\Sigma}^{(0)*})^{-1}\|_2 = o(t^{-1/2})$ and $\|\mathbf{\Sigma}^{-1}(t)\|_2 = O(1)$. Under the given conditions, equation 21 and the first assertion in equation 17 now follow from the arguments used in Part 2 and 3 of the proof of Lemma E.2, Shi et al. (2020a).

### F.6 PROOF OF LEMMA 4

The asymptotic normality of $\sqrt{t}\{\hat{\tau}(t) - \tau_0\}/\sigma(t)$ can be proven using similar arguments in Part 3 of the proof of Theorem 3. In the following, we focus on equation 24. Define the matrix

$$\mathbf{\Omega}^{(0)*} = \int_{s\in\mathbb{S}} \sum_{a\in\{0,1\}} \mathrm{E}\left\{\left(\begin{array}{c}\xi_0\varepsilon_{0,0}\\\xi_0\varepsilon_{0,1}\end{array}\right)\left(\begin{array}{c}\xi_0\varepsilon_{0,0}\\\xi_0\varepsilon_{0,1}\end{array}\right)^\top\middle| S_0 = s, A_0 = a\right\}\mu(s)\{ab^{(0)}(s) + (1-a)(1 - b^{(0)}(s))\}ds.$$

Similar to Lemma E.5 of Shi et al. (2020a), we can show $\|\mathbf{\Omega}^{(0)*} - \mathbf{\Omega}(t)\|_2 = O(t^{-1/2})$. Thus, it suffices to show $\inf_q \lambda_{\min}(\mathbf{\Omega}^{(0)*}) > 0$. Under CA and SRA, we have

$$\mathrm{E}\left\{\left(\begin{array}{c}\xi_0\varepsilon_{0,0}\\\xi_0\varepsilon_{0,1}\end{array}\right)\left(\begin{array}{c}\xi_0\varepsilon_{0,0}\\\xi_0\varepsilon_{0,1}\end{array}\right)^\top\middle| S_0 = s, A_0 = a\right\}$$

$$= \mathrm{E}\left\{\left(\begin{array}{c}\xi_0(a)\varepsilon^*(0,a)\\\xi_0(a)\varepsilon^*(1,a)\end{array}\right)\left(\begin{array}{c}\xi_0(a)\varepsilon^*(0,a)\\\xi_0(a)\varepsilon^*(1,a)\end{array}\right)^\top\middle| S_0 = s, A_0 = a\right\}$$

$$= \mathrm{E}\left\{\left(\begin{array}{c}\xi_0(a)\varepsilon^*(0,a)\\\xi_0(a)\varepsilon^*(1,a)\end{array}\right)\left(\begin{array}{c}\xi_0(a)\varepsilon^*(0,a)\\\xi_0(a)\varepsilon^*(1,a)\end{array}\right)^\top\middle| S_0 = s\right\}$$

For any $2q$-dimensional vectors $\boldsymbol{a}_1, \boldsymbol{a}_2$ that satisfy $\|\boldsymbol{a}_1\|_2^2 + \|\boldsymbol{a}_2\|^2 = 1$, it follows that

$$(\boldsymbol{a}_1^\top, \boldsymbol{a}_2^\top)\mathrm{E}\left\{\left(\begin{array}{c}\xi_0\varepsilon_{0,0}\\\xi_0\varepsilon_{0,1}\end{array}\right)\left(\begin{array}{c}\xi_0\varepsilon_{0,0}\\\xi_0\varepsilon_{0,1}\end{array}\right)^\top\middle| S_0 = s, A_0 = a\right\}(\boldsymbol{a}_1^\top, \boldsymbol{a}_2^\top)^\top$$

$$= \{\boldsymbol{a}_1^\top\xi(s,a)\}^2\mathrm{E}[\{\varepsilon^*(0,a)\}^2|S_0 = s] + \{\boldsymbol{a}_2^\top\xi(s,a)\}^2\mathrm{E}[\{\varepsilon^*(1,a)\}^2|S_0 = s]$$

$$+ 2\{\boldsymbol{a}_1^\top\xi(s,a)\}\{\boldsymbol{a}_2^\top\xi(s,a)\}\mathrm{E}[\{\varepsilon^*(0,a)\varepsilon^*(1,a)\}|S_0 = s]$$

$$\geq \{\boldsymbol{a}_1^\top\xi(s,a)\}^2\mathrm{E}[\{\varepsilon^*(0,a)\}^2|S_0 = s] + \{\boldsymbol{a}_2^\top\xi(s,a)\}^2\mathrm{E}[\{\varepsilon^*(1,a)\}^2|S_0 = s]$$

$$- 2\rho_\varepsilon|\boldsymbol{a}_1^\top\xi(s,a)||\boldsymbol{a}_2^\top\xi(s,a)|\sqrt{\mathrm{E}[\{\varepsilon^*(0,a)\}^2|S_0 = s]\mathrm{E}[\{\varepsilon^*(1,a)\}^2|S_0 = s]}$$

$$= (1 - \rho_\varepsilon)\{\boldsymbol{a}_1^\top\xi(s,a)\}^2\mathrm{E}[\{\varepsilon^*(0,a)\}^2|S_0 = s] + (1 - \rho_\varepsilon)\{\boldsymbol{a}_2^\top\xi(s,a)\}^2\mathrm{E}[\{\varepsilon^*(1,a)\}^2|S_0 = s]$$

$$+ \rho_\varepsilon\left||\boldsymbol{a}_1^\top\xi(s,a)|\sqrt{\mathrm{E}[\{\varepsilon^*(0,a)\}^2|S_0 = s]} + |\boldsymbol{a}_2^\top\xi(s,a)|\sqrt{\mathrm{E}[\{\varepsilon^*(1,a)\}^2|S_0 = s]}\right|^2$$

$$\geq (1 - \rho_\varepsilon)\{\boldsymbol{a}_1^\top\xi(s,a)\}^2\mathrm{E}[\{\varepsilon^*(0,a)\}^2|S_0 = s] + (1 - \rho_\varepsilon)\{\boldsymbol{a}_2^\top\xi(s,a)\}^2\mathrm{E}[\{\varepsilon^*(1,a)\}^2|S_0 = s],$$

where $\rho_\varepsilon = \sup_q \sup_{a \in \{0,1\}, s \in \mathbb{S}} \rho_\varepsilon(a, s)$. Under C3, we have $\rho_\varepsilon < 1$ and $\inf_q \inf_{a',a,s} \mathrm{E}[\{\varepsilon^*(a', a)\}^2 | S_0 = s] > 0$. It follows that

$$(\boldsymbol{a}_1^\top, \boldsymbol{a}_2^\top) \mathrm{E} \left\{ \begin{pmatrix} \xi_0 \varepsilon_{0,0} \\ \xi_0 \varepsilon_{0,1} \end{pmatrix} \begin{pmatrix} \xi_0 \varepsilon_{0,0} \\ \xi_0 \varepsilon_{0,1} \end{pmatrix}^\top \middle| S_0 = s, A_0 = a \right\} (\boldsymbol{a}_1^\top, \boldsymbol{a}_2^\top)^\top \geq \bar{c}[\{\boldsymbol{a}_1^\top \xi(s, a)\}^2 + \{\boldsymbol{a}_2^\top \xi(s, a)\}^2],$$

for some constant $\bar{c}_3 > 0$. Therefore,

$$\lambda_{\min}(\boldsymbol{\Omega}^{(0)*}) = \inf_{\|\boldsymbol{a}_1\|_2^2 + \|\boldsymbol{a}_2\|_2^2 = 1} (\boldsymbol{a}_1^\top, \boldsymbol{a}_2^\top) \boldsymbol{\Omega}^{(0)*} (\boldsymbol{a}_1^\top, \boldsymbol{a}_2^\top)^\top$$

$$\geq \bar{c}_3 \inf_{\|\boldsymbol{a}_1\|_2^2 + \|\boldsymbol{a}_2\|_2^2 = 1} \int_{s \in \mathbb{S}} \sum_{a \in \{0,1\}} [\{\boldsymbol{a}_1^\top \xi(s, a)\}^2 + \{\boldsymbol{a}_2^\top \xi(s, a)\}^2] \mu(s) \{ab^{(0)}(s) + (1 - a)(1 - b^{(0)}(s))\} ds.$$

The strict positivity of $\mu(\cdot)$ and the condition that $b^{(0)}(\cdot)$ is uniformly bounded away from 0 and 1 yields

$$\lambda_{\min}(\boldsymbol{\Omega}^{(0)*}) \geq \bar{c}_4 \inf_{\|\boldsymbol{a}_1\|_2^2 + \|\boldsymbol{a}_2\|_2^2 = 1} \int_{s \in \mathbb{S}} \sum_{a \in \{0,1\}} [\{\boldsymbol{a}_1^\top \xi(s, a)\}^2 + \{\boldsymbol{a}_2^\top \xi(s, a)\}^2] ds, \tag{42}$$

for some constant $\bar{c}_4 > 0$. With some calculation, we can show the RHS of equation 42 is equal to

$$\bar{c}_4 \lambda_{\min} \left\{ \int_{s \in \mathbb{S}} \Psi(s) \Psi^\top(s) \right\}.$$

By Condition C2(ii), it is strictly positive. This yields $\inf_q \lambda_{\min}(\boldsymbol{\Omega}^{(0)*}) > 0$. Thus, we obtain equation 24.

### F.7 PROOF OF LEMMA 5

We begin by proving

$$\|\boldsymbol{\Sigma}^{-1}(t)\|_2 = O(1), \tag{43}$$

under D2. For any matrices $\boldsymbol{M}_1$ and $\boldsymbol{M}_2$, denote by $\mathrm{diag}[\boldsymbol{M}_1, \boldsymbol{M}_2]$ the block diagonal matrix

$$\begin{bmatrix} \boldsymbol{M}_1 & \\ & \boldsymbol{M}_2 \end{bmatrix}.$$

By MA and Condition C1(ii), the two Markov chains $\{S_{2t-1}\}_{t \geq 1}$, $\{S_{2t}\}_{t \geq 0}$ are geometrically ergodic. Let $\mu_1$ and $\mu_2$ denote the density function of their stationary distributions, respectively. Under C1(i), we can similarly show that they are uniformly bounded away from 0 and $\infty$. Define

$$\boldsymbol{\Sigma}_1^* = \int_{s,s' \in \mathbb{S}} \mathrm{diag}[\xi(s, 1)\{\xi(s, 1) - \gamma \xi(s', 0)\}^\top, \xi(s, 1)\{\xi(s, 1) - \gamma \xi(s', 1)\}^\top] \mu_1(s) p(s'; 1, s) ds ds',$$

$$\boldsymbol{\Sigma}_2^* = \int_{s,s' \in \mathbb{S}} \mathrm{diag}[\xi(s, 0)\{\xi(s, 0) - \gamma \xi(s', 0)\}^\top, \xi(s, 0)\{\xi(s, 0) - \gamma \xi(s', 1)\}^\top] \mu_2(s) p(s'; 0, s) ds ds'.$$

The matrix $(\boldsymbol{\Sigma}_1^* + \boldsymbol{\Sigma}_2^*)/2$ corresponds to the population limit of $\boldsymbol{\Sigma}(t)$. Similar to Lemma E.5 of Shi et al. (2020a), we can show $\|\boldsymbol{\Sigma}_1^* - (2t)^{-1} \sum_{j=0}^t \mathrm{E} \boldsymbol{\Sigma}_{2j+1}\|_2 = o(t^{-1/2})$ and $\|\boldsymbol{\Sigma}_2^* - (2t)^{-1} \sum_{j=0}^t \mathrm{E} \boldsymbol{\Sigma}_{2j}\|_2 = o(t^{-1/2})$. This further yields

$$\left\| \frac{\boldsymbol{\Sigma}_1^* + \boldsymbol{\Sigma}_2^*}{2} - \boldsymbol{\Sigma}(t) \right\|_2 = o(t^{-1/2}).$$

Similar to the proof of Lemma 3, in order to show equation 43, it suffices to show

$$\|(\boldsymbol{\Sigma}_1^* + \boldsymbol{\Sigma}_2^*)^{-1}\|_2 = O(1). \tag{44}$$

Notice that $\boldsymbol{\Sigma}_1^* + \boldsymbol{\Sigma}_2^* = \mathrm{diag}[\boldsymbol{\Sigma}^*(0), \boldsymbol{\Sigma}^*(1)]$ where

$$\boldsymbol{\Sigma}^*(a) = \int_{s,s' \in \mathbb{S}} [\xi(s, 0)\{\xi(s, 0) - \gamma \xi(s', a)\} \mu_2(s) p(s'; 0, s) + \xi(s, 1)\{\xi(s, 1) - \gamma \xi(s', a)\} \mu_1(s) p(s'; 1, s)] ds ds'.$$

The matrix $\boldsymbol{\Sigma}^*(0)$ can be further decomposed into

$$\boldsymbol{\Sigma}^*(0) = \left[ \begin{array}{cc} \boldsymbol{\Sigma}_{1,1}^*(0) & \\ \boldsymbol{\Sigma}_{2,1}^*(0) & \boldsymbol{\Sigma}_{2,2}^*(0) \end{array} \right],$$

where

$$\boldsymbol{\Sigma}_{1,1}^*(0) = \int_{s,s'\in\mathbb{S}} \Psi(s)\{\Psi(s) - \gamma\Psi(s')\}^\top \mu_2(s)p(s';0,s)dsds',$$

$$\boldsymbol{\Sigma}_{2,1}^*(0) = -\gamma \int_{s,s'\in\mathbb{S}} \Psi(s)\Psi^\top(s')\mu_1(s)p(s';1,s)dsds',$$

$$\boldsymbol{\Sigma}_{2,2}^*(0) = \int_s \Psi(s)\Psi^\top(s)\mu_1(s)ds.$$

Similar to the proof of Lemma 3, we can show $\|\{\boldsymbol{\Sigma}_{1,1}^*(0)\}^{-1}\|_2 = O(1)$, $\|\{\boldsymbol{\Sigma}_{2,2}^*(0)\}^{-1}\|_2 = O(1)$ and $\|\boldsymbol{\Sigma}_{2,1}^*(0)\|_2 = O(1)$. It follows that $\|\{\boldsymbol{\Sigma}^*(0)\}^{-1}\|_2 = O(1)$. Similarly, we can show $\|\{\boldsymbol{\Sigma}^*(1)\}^{-1}\|_2 = O(1)$. This proves equation 44. Thus, we obtain equation 43.

Using similar arguments in Part 2 of the proof of Lemma E.2, Shi et al. (2020a), we can show $\|t^{-1}\sum_{j=0}^{t-1}\boldsymbol{\Sigma}_{2j} - \boldsymbol{\Sigma}_2^*\|_2 = O_p(t^{-1/2}\log t)$ and $\|t^{-1}\sum_{j=0}^{t-1}\boldsymbol{\Sigma}_{2j+1} - \boldsymbol{\Sigma}_1^*\|_2 = O_p(t^{-1/2}\log t)$. This further implies $\|\widehat{\boldsymbol{\Sigma}}(t) - (\boldsymbol{\Sigma}_1^* + \boldsymbol{\Sigma}_2^*)/2\| = O_p(t^{-1/2}\log t)$ and hence $\|\widehat{\boldsymbol{\Sigma}}(t) - \boldsymbol{\Sigma}(t)\|_2 = O_p(t^{-1/2}\log t)$. Combining these results together with equation 43, we can show equation 21 and the first assertion in equation 17 hold. equation 20 and the second assertion in equation 17 hold can be proven in a similar manner.

Finally, using similar arguments in the proof of Lemma 4, we can show equation 24 holds. We omit the details to save space.

### F.8 PROOF OF LEMMA 6

Under C1(iv), we have equation 6 holds. Similar to equation 7, we can show $\Pi^{(k)}$ has a probability density function $\mu^{(k)}$ given by

$$\mu^{(k)}(s') = \sum_{a\in\{0,1\}} \int_{s\in\mathbb{S}} [a\{1 - b^{(k)}(s)\} + (1-a)b^{(k)}(s)]p(s';a,s)\Pi^{(k)}(ds). \tag{45}$$

For $a' \in \{0,1\}$, define

$$\boldsymbol{\Sigma}^{(k)*}(a) = \int_{s,s'\in\mathbb{S}} \sum_{a\in\{0,1\}} \xi(s,a)\{\xi(s,a) - \gamma\xi(s',a')\}^\top \mu^{(k)}(s)\{a\{1-b^{(k)}(s)\} + (1-a)b^{(k)}(s)\}p(s';a,s)dsds'.$$

Condition on $\{(S_j, A_j, Y_j)\}_{1\leq j<T_{k-1}}$, the matrix $\boldsymbol{\Sigma}^{(k)*}(a)$ is deterministic. Let $\boldsymbol{\Sigma}^{(k)} = \text{diag}[\boldsymbol{\Sigma}^{(k)*}(0), \boldsymbol{\Sigma}^{(k)*}(1)]$. Similar to the proof of Lemma 3, we can show $\|\boldsymbol{\Sigma}^{(k)*} - \boldsymbol{\Sigma}^{(k)}\|_2 = o(1)$, conditional on $\{(S_j, A_j, Y_j)\}_{1\leq j<T_{k-1}}$, with probability tending to 1. This implies for any sufficiently small $\epsilon > 0$,

$$\text{Pr}(\|\boldsymbol{\Sigma}^{(k)*} - \boldsymbol{\Sigma}^{(k)}\|_2 > \epsilon|\{(S_j, A_j, Y_j)\}_{1\leq j<T_{k-1}}) \xrightarrow{P} 0.$$

The above conditional probability is bounded between 0 and 1. Using bounded convergence theorem, we have

$$\text{Pr}(\|\boldsymbol{\Sigma}^{(k)*} - \boldsymbol{\Sigma}^{(k)}\|_2 > \epsilon) = o(1), \tag{46}$$

and hence $\|\boldsymbol{\Sigma}^{(k)*} - \boldsymbol{\Sigma}^{(k)}\|_2 = o_p(1)$.

Notice that $\sup_s |b^{(k)}(s) - b^*(s)| \xrightarrow{P} 0$ and $\|\Pi^{(k)} - \Pi^*\|_{TV} \xrightarrow{P} 0$. Define

$$\mu^*(s) = \sum_{a\in\{0,1\}} \int_{s\in\mathbb{S}} [a\{1 - b^*(s)\} + (1-a)b^*(s)]p(s';a,s)\Pi^*(ds).$$

It follows that

$$|\mu^{(k)}(s') - \mu^*(s')| \leq \sum_{a \in \{0,1\}} \int_{s \in \mathbb{S}} \{a|b^*(s) - b^{(k)}(s)| + (1-a)|b^*(s) - b^{(k)}(s)|\} p(s'; a, s) \Pi^{(k)}(ds)$$

$$+ \sum_{a \in \{0,1\}} \int_{s \in \mathbb{S}} [a\{1 - b^*(s)\} + (1-a)b^*(s)] p(s'; a, s) |\Pi^{(k)}(ds) - \Pi^*(ds)|,$$

and hence $\sup_s |\mu^{(k)}(s) - \mu^*(s)| \xrightarrow{P} 0$. With some calculations, we can show $\|\Sigma^{(k)*}(a) - \Sigma^*(a)\|_2 \xrightarrow{P} 0$ where

$$\Sigma^*(a) = \int_{s, s' \in \mathbb{S}} \sum_{a \in \{0,1\}} \xi(s, a) \{\xi(s, a) - \gamma\xi(s', a')\}^\top \mu^*(s) \{a\{1 - b^*(s)\} + (1-a)b^*(s)\} p(s'; a, s) ds ds'.$$

Let $\Sigma^* = \text{diag}[\Sigma^*(0), \Sigma^*(1)]$, we obtain $\|\Sigma^{(k)*} - \Sigma^*\|_2 = o_p(1)$. Combining this together with equation 46, we obtain $\|\Sigma^{(k)} - \Sigma^*\|_2 = o_p(1)$. The proof of Lemma 3 yields $\|\Sigma^{(1)} - \Sigma^{(0)*}\|_2 = o(1)$. Thus, we have for any $2 \leq k \leq K$ that

$$\|\overline{\Sigma}^{(k)} - T_k^{-1}T_1\Sigma^{(0)*} - T_k^{-1}(T_k - T_1)\Sigma^*\|_2 = o_p(1). \tag{47}$$

Similar to the proof of Lemma 3, we can show $\mu^{(k)}$'s are uniformly bounded away from 0 and $\infty$. It follows that $\mu^*$ is uniformly bounded away from 0 and $\infty$. Using similar arguments in Lemma 3, we can show $\|\{T_k^{-1}T_1\Sigma^{(0)*} + T_k^{-1}(T_k - T_1)\Sigma^*\}^{-1}\|_2 = O(1)$. Using similar arguments in Part 1 of the proof of Lemma E.2, Shi et al. (2020a), we have by equation 47 that $\|(\overline{\Sigma}^{(k)})^{-1}\|_2 = O(1)$, with probability tending to 1. equation 32 is thus proven.

Assertion equation 31 now follows using similar arguments in Part 2 and Part 3 of the proof of Lemma E.2, Shi et al. (2020a).

## G ADDITIONAL FIGURES

We present some additional figures to report the simulation results in this section. Figure 4 depicts the empirical rejection probabilities of the modified version of the O'Brien & Fleming sequential test developed by Kharitonov et al. (2015). It can been seen that such a test has no power at all. In addition, we remark that Kharitonov et al. (2015)'s test requires equal sample size $T_1 = T_k - T_{k-1}$ for $k = 2, \cdots, K$ and is not directly applicable to our setting with unequal sample size. To apply such a test, we modify the decision time and set $(T_1, T_2, T_3, T_4, T_5) = (120, 240, 360, 480, 600)$.

Figure 5 depicts the empirical rejection probabilities of our test and two-sample t-test with the error spending function given by $\alpha_2$. Figure 6 reports the empirical rejection probabilities of our test with different combinations of the number of basis and the error spending function.

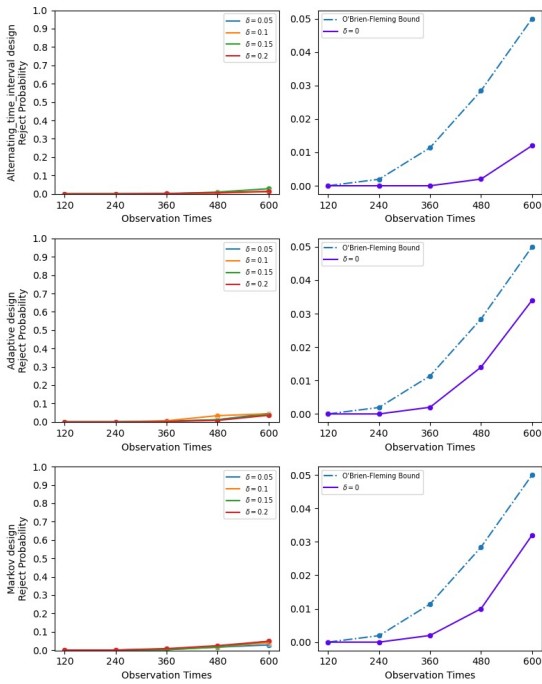

Figure 4: Empirical rejection probabilities of the modified version of the O'Brien & Fleming sequential test developed by Kharitonov et al. (2015). The left panels depicts the empirical type-I error and the right panels depicts the empirical power. Settings correspond to the alternating-time-interval, adaptive and Markov design, from top plots to bottom plots.

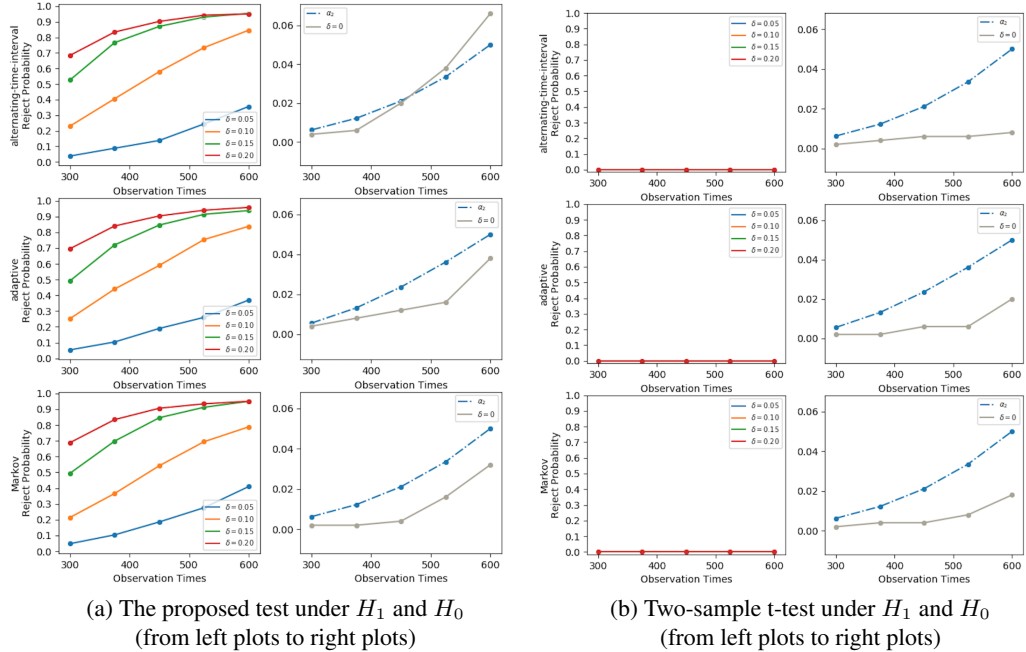

(a) The proposed test under $H_1$ and $H_0$ (from left plots to right plots)

(b) Two-sample t-test under $H_1$ and $H_0$ (from left plots to right plots)

Figure 5: Empirical rejection probabilities of our test and the two-sample t-test with $\alpha(\cdot) = \alpha_2(\cdot)$. Settings correspond to the alternating-time-interval, adaptive and Markov design, from top plots to bottom plots.

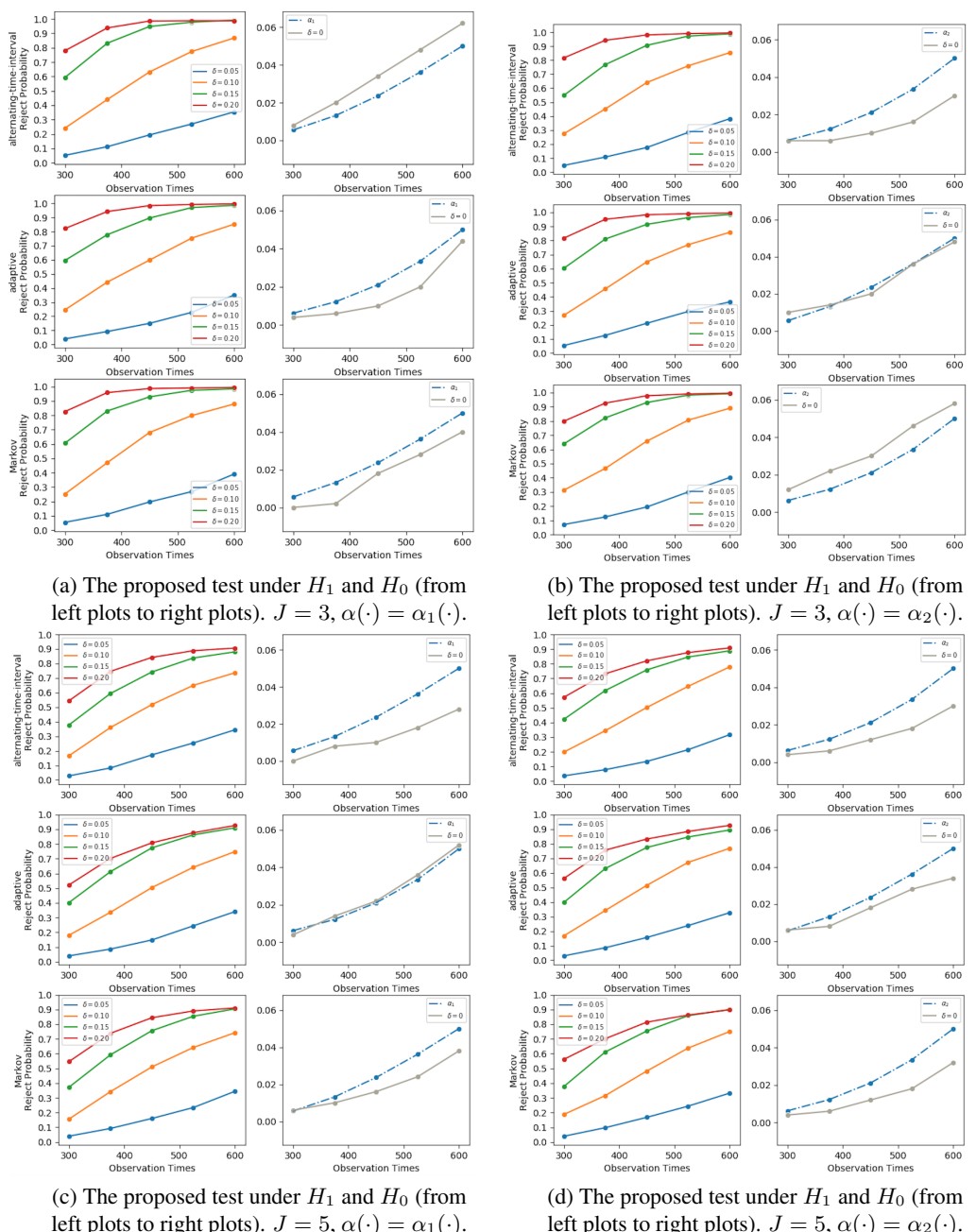

(a) The proposed test under $H_1$ and $H_0$ (from left plots to right plots). $J = 3, \alpha(\cdot) = \alpha_1(\cdot)$.

(b) The proposed test under $H_1$ and $H_0$ (from left plots to right plots). $J = 3, \alpha(\cdot) = \alpha_2(\cdot)$.

(c) The proposed test under $H_1$ and $H_0$ (from left plots to right plots). $J = 5, \alpha(\cdot) = \alpha_1(\cdot)$.

(d) The proposed test under $H_1$ and $H_0$ (from left plots to right plots). $J = 5, \alpha(\cdot) = \alpha_2(\cdot)$.

Figure 6: Empirical rejection probabilities of our test. Settings correspond to the alternating-time-interval, adaptive and Markov design, from top plots to bottom plots.

