# OpenReview forum: "A REINFORCEMENT LEARNING FRAMEWORK FOR TIME DEPENDENT CAUSAL EFFECTS EVALUATION IN A/B TESTING"
_ICLR.cc/2021/Conference — Reject_

### Official Review · AnonReviewer4 · 2020-10-19
**Official Blind Review #4**

**Rating:** 6
**Confidence:** 2

**Review:**

*Summary*

This paper introduces a reinforcement learning framework for testing the difference in long-term treatment effects between treatment and control in online experiments. The proposed testing procedure allows for sequential monitoring and online update, has good control of the type I error rate at interim time points, and has non-negligible powers against local alternatives. Its performance is tested on both synthetic data sets and a real-world ride-sharing data set.

*Assessment*

I'm not too familiar with literature but slightly leaning towards acceptance since I find the subject somewhat interesting. But still I want more information regarding novelty to make the final decision. Also the technical proofs are quite long and unfortunately I do not have time to go through all of them.

*Pros*

- The problem considered in the paper seems relevant in practice.

- The algorithm is generally clear and intuitive.

- Empirical performance of the algorithm looks overall good.

*Cons and Questions*

- The proposed $\hat{\tau}(t)$ looks like the difference between two OPE estimators using direct method that requires $o(T^{-1/2})$ approximation accuracy. There are other OPE estimators that have asymptotic distribution guarantees, e.g., Kallus and Uehara (2019) as is cited in the paper (though they require the propensity to be bounded away from 0 and 1). If we consider a Markov design setting where the bounded propensity assumption does hold true, can these estimators be directly applied to do sequential online testing? It would be desirable to also compare with them in synthetic studies.

- Several key notations are delayed to the appendix, which makes the reading experience not very smooth.


*Minor Comments and Typos*

- The initial value for $S_0$ in synthetic analysis seems to be missing.

- Page 2 Line 5: "to allows" $\rightarrow$ "to allow"

- Last line, 2nd paragraph in section 2: "these method" $\rightarrow$ "these methods"

---

> ### Author Response · Authors · 2020-11-22
> **Response to Reviewer 4**
>
> We greatly thank your valuable comments, many of which will lead to a more readable and self-contained version of our paper. We attempt to address all the points in the following. The revised manuscript taking into accounts all your suggestions has been uploaded. Please refer to the most updated revision for details.
>
> **Other OPE estimators for sequential online testing**
>
> Thanks for this excellent suggestion. Other OPE estimators cannot be directly applied. However, it could be coupled with our proposed Bootstrap procedure for sequential online testing. Following your suggestion, we use the double reinforcement learning method (DRL, Kallus and Uehara, 2019) as an example for detailed illustration.
>
> First, we remark that DRL requires the system to be ergodic and use an inverse propensity-score weighted method to construct the value estimator. As such, it might not be applicable to the alternating-time-interval design and the adaptive design.
>
> Second, in the Markov design, it could be coupled with our bootstrap procedure for sequential testing. We compare such a procedure with our proposed method using the simulation setting in Section 5.1 and report the rejection probabilities in Figure 3 (page 13). It can be seen that DRL-based test has some inflated type-I errors under $H_0$ and is less powerful than our procedure under $H_1$.
>
> Please refer to Appendix B.2 (page 13) for details.
>
> **Several key notations are delayed to the appendix**
>
> Thanks for this comment, we have moved several notations in the main context to make it more self-contained. For instance, the definition of the matrix $U$ is now moved to Section 4.2 (page 4, line-21). For some other notations, we decide to keep their definitions in appendix to save space and make the main text less dense. However, we do add some informal definitions to explain these notations. For instance, the explicit form of the estimator $\widehat{\Psi}$ is introduced in the appendix. In the main text, we explain that it corresponds to some consistent covariance estimator of $\eta_t$ based on the data observed at time $t$ (page 5, line-20).
>
> **Minor comments and typos**
>
> We apologize for the typos. They have been corrected. As for the initial states, we have added their distributions in the paper (page 7, line 10). Thanks for pointing these out.
>
> We once again appreciate your effort on reviewing our paper. We hope that the above discussion has addressed your comments.

---

### Official Review · AnonReviewer2 · 2020-10-26
**The paper applies the reinforcement learning (RL) framework to A/B test on scenarios that have a sequential characteristic on state and reward transitions.**

**Rating:** 5
**Confidence:** 2

**Review:**

The paper considers the A/B testing scenario where there are a sequence of actions that would change the environment state and the reward, such as the ride-sharing dispatching. Moreover, the authors formulate the RL framework in terms of the potential outcome rather than by observed data, since they want to model the off-policy evaluation scenario where only the observed data are collected. The authors propose the conditions for the indentifiability of the hypothesis testing, propose a testing procedure, provide its theoretical analysis on type-I error and power, and conduct experiments on both synthetic data and real-world data.

The writing of the paper is too dense, and thus it severely impacts the understanding of the technical content of the paper. One issue is that the main text of the paper intensely refer to contents in the appendix, including the actual testing procedure pseudocode, the discussions on many aspects of their study, and many additional results. The appendix is not required to be read by the reviewers, but the current text without these appendix is not self-contained. Another issue is that intuitive explanation and discussion are not enough. For example, Lemma 1 is the key lemma for the identifiability result, but how to interpret it? Why does it mean identifiability? How does it depend on the four assumptions? There is lack of conceptual explanation of this lemma. Theorem 3 (type-I error) and Theorem 4 (power) also lack discussions. Moreover, three treatment designs D1, D2, and D3 in Section 4.2 also lack discussions. Why do we consider these three cases? Are they general enough to cover many application scenarios? What about other cases? Overall, I feel that the main issue of the paper is that its technical part is too dense, lacking many needed explanations. As a result, it is hard to appreciate the novelty of the paper, and it is unclear what is the exact new contribution of the paper.

In contrast, the rea-world data evaluation from the ride-sharing dispatching is the most informative part I found in the paper. Through this test, the authors demonstrate that the standard t-test fails to discover the carry over effect and cannot distinguish the two policies, while the test proposed by the authors could detect the long-term effect and show the advantage of the new policy. I hope that the authors could elaborate more on this real-world test, and connect it with their theoretical analysis to explain how their test could improve upon the standard t-test.

Minor comments:

- P15, Lemma 2
The second Q(a',a',s'). Should it be Q(a', a, s')?

---

> ### Author Response · Authors · 2020-11-22
> **Response to Reviewer 2**
>
> We greatly thank your valuable comments, many of which will lead to a more readable and self-contained version of our paper. We attempt to address all the points in the following. The revised manuscript taking into accounts all your suggestions has been uploaded. Please refer to the most updated revision for details.
>
> **The writing is too dense, the main context intensively refers to the content in the appendix**
>
> Thanks for this excellent comment!
>
> First, to improve the readability, we move some extra technical results to the Appendix. For instance, we move Theorem 3 (Theorem 1 in our submitted manuscript) that derives the asymptotic normality of our test statistics to Appendix D (page 15).
>
> Second, we move many necessary contents in the appendix to the main text to make it more self-contained. For instance, the definition of the vector $U$ (page 5, line-21) is introduced in the main text now. We also add inline sketch of proofs for Lemma 1, Theorems 1 and 2 (page 4 and 6) and related discussions for these propositions as well as discussions of D1-D3 (page 6, Section 4.4) to make the main text more self-contained.
>
> Third, we explain some dependent concepts in the paper. For instance, we have added a self-contained definition of the $\alpha$-spending function approach in the paper (page 2, line-20 to line-18).
>
> We hope these could help improve the readability of the paper.
>
> **Add Intuitive Explanation and Discussions**
>
> Thanks for this excellent suggestion! We have added the related discussions in the paper.
>
> For D1-D3, we added following discussions (page 6, Section 4.4):
>
> “Here, D2 is a deterministic design and is widely used in industry (see our real data example). D1 and D3 are random designs. D1 is commonly assumed in the literature on off-policy evaluation (Jiang and Li, 2016). D3 is widely employed in the contextual bandit setting to balance the trade-off between exploration and exploitation. These three settings cover a variety of scenarios in practice.”
>
> For lemma 1, the identifiability of $\tau_0$ means that $\tau_0$ is estimable from the observed data. We add an inline sketch of proof for Lemma 1. As discussed in the sketch of the proof, the four conditions MA, CMIA, CA and SRA implies that the Q-function defined under the potential outcome framework is the same as that defined on the observed data. In addition, the following discussions have been added (page 4):
>
> “Lemma 1 implies that the Q-function is estimable from the observed data. Specifically, an estimating equation can be constructed based on Lemma 1 and the Q-function can be learned by solving this estimating equation. Note that $V(a,s)=Q(a;a,s)$ and $\tau_0$ is completely determined by the value function $V$. Notice that $V(a;s)=Q(a;a,s)$ for any $a$ and $s$.
> As a result, $\tau_0$ is estimable from the observed data as well}.”
>
> For Theorem 1, we added the following discussions (page 6):
>
> “Theorem 1 implies that the type-I error rate of the proposed test is well controlled. When ATE$=0$, the equality in Theorem 1 holds. The rejection probability achieves the nominal level under $H_0$.”
>
> For Theorem 2, we added the following discussions (page 6):
>
> “The second assertion in Theorem 2 implies that our test has non-negligible powers against local alternatives converging to $H_0$ at the $T^{-1/2}$ rate. When the signal decays to zero faster than this rate, our test is not able to detect $H_1$. When the signal decays at a slower rate, the power of our test approaches 1. Combining Theorems 1 with 2 yields the consistency of our test.”
>
> We also add in-line sketch of proofs for these two theorems.
>
> Finally, we add a toy example in Section 4.1 (page 4) to intuitively demonstrate the limitations of existing causal inference and A/B testing methods.
>
> **Elaborate more on this real-world test and connect it with their theoretical analysis**
>
> Thanks for this comment. Following your suggestion, we added the following discussions (page 8, line-5):
>
> This result is consistent with our findings. Specifically, the treatment effect at a given time affects the distribution of drivers in the future, inducing interference in time. As shown in the toy example (see Section 4.1), the t-test cannot detect such carryover effects, leading to a low power. Our procedure, according to Theorem 2, has enough powers to discriminate $H_1$ from $H_0$.
>
> **Minor Comments**
>
> It should be $Q(a’;a’,s’)$. According to definition, we have $Q(a’;a’,s’)=V(a’;s’)$ where $V$ is the value function. Consequently, we have $Q(a’;a,s)=r(a,s)+\gamma \int_{s’}V(a’;s)\mathcal{P}(ds’;a,s)$. This is consistent with the Bellman equation for the Q-function (see e.g., Equation (4.6) in Sutton and Barto, 2018).
>
> We once again appreciate your effort in reviewing our paper. We hope that the above discussion can address your comments.

---

### Official Review · AnonReviewer3 · 2020-10-26
**A creative proposition, but is the RL framework necessary ?**

**Rating:** 5
**Confidence:** 4

**Review:**

Summary: the paper is motivated by the problem of sequential experimentation as found in online experiments; more precisely the authors propose to use an RL framework to measure performance while optimizing each treatment policy during experiment. The proposal gives rises to a sequential statistical test procedure to decide at the earliest which treatment policy is significantly better than the others. Theoretically, the power and precision of the sequential decision process are studied and proven to be non trivially efficient. Experiments showcase the method on synthetic and private datasets, including influence of the true effect size on test trigger.

Good points:
- [novelty] the rewriting of Q/V functions as counterfactuals seems quite original
- [impact] the problem of early termination and optimization of online experiment decision is important for many industries
- [scientific rigor] detailed and rigorous statement of setting and assumptions
- [impact] proofs of the test precision and power give important guarantees

Questions:
- is the treatment policy $\pi$ deterministic ? I'm confused as it is in Sec. 3.1 but not in the comment after Lemma 1 (also "certain conditions" remains mysterious to me)
- Why "in our setup, test statistics no longer have the canonical joint distribution" (Sec. 2) - could you elaborate on that ?

 Points currently limiting the relevance of the paper:
- Current write-up falls short in convincing me that an RL framework is indeed necessary to solve sequential testing / early stopping of experiments. I would have appreciated a simple example showcasing the shortcomings of existing approaches (beyond simple t-test) to fix ideas and identify critical properties of a testing strategy that solves the problem.
- I would have expected more discussion of early stopping methods for sequential testing, e.g. https://dl.acm.org/doi/abs/10.1145/2766462.2767729 . As the problem is pervasive for tech companies it would be very surprising that no competing methods exist. Even if they don't have all the theoretical properties of the one proposed I suspect they would still be interesting to compare in experiments. Oftentimes even invalid methods show competitive behavior in practice.
- I'm in doubt whether the most relevant baselines were used in the experiments. Especially for the ride-sharing system I believe using other baselines than just t-test would highlight the benefits of the proposed test. Regarding the t-test that is used: I believe using it at different times in the experiment as seen in Fig. 2 jeopardizes any guarantees without adjustment for multiple testing. If Bonferroni was used to correct it it is thus quite understandable why it fails dramatically. The work of Abhishek & Manor 2017 cited as related works would have been a much better baseline in that respect (but other alternatives exist). Moreover, I would have expected to see more advanced techniques that optimize the decision time as baselines as this problem is quite old and pervasive now for many tech industries such as search, advertising and so on.
- I would be interested in the point of view of authors whether the bandit literature on sequential A/B testing could be relevant here, e.g. https://arxiv.org/abs/1905.11797 or https://arxiv.org/abs/1905.11797
- Propositions would have benefited to have a sketch of the proof inline
- Some dependent concepts (e.g. $\alpha$-spending) are used without a self-contained definition or reader friendly introduction

Points that could improve the paper (and my score):
- add an example showcasing the shortcomings of existing approaches; starting probably with the t-test and ending with more advanced baselines (see below)
- add discussion of more "early stopping" baselines (e.g. https://ieeexplore.ieee.org/abstract/document/7796910 + refs above)
- add more relevant baselines in experiments to clarify impact of the contribution w.r.t. existing practice

Overall my feeling for now is that the proposed technique is quite original and promising but that it lacks strong arguments to show that it is the simplest solution giving as good performance. I'm quite inclined to review my score based on the forthcoming discussion.

---

> ### Author Response · Authors · 2020-11-22
> **Response to Reviewer 3 Part 2**
>
> **Point 3 and Q4, Q5, More relevant baselines in the experiments** Again, many thanks for this excellent comment! Following your suggestion, we have added several baselines in the experiments.
>
> First, to implement different early stopping methods, in our synthetic data example (Section 5.1, page 7, line 5), we compare our procedure with a modified version of the O’Brien & Fleming’s test developed by Kharitonov et al. (2015). As we discussed in our toy example (Section 4.1), such a test cannot detect carryover effects in time. Consequently, similar to the t-test, it does not have any power at all (see Figure 4 on page 32). We remark that this is the limitation of existing A/B tests. No matter how they optimize the decision time, they are powerless if they cannot detect the carryover effects.
>
> Second, to implement different causal inference methods, in our toy example (Section 4.1, page-10), we compare our test with a test based on the double machine learning method (DML, Chernozhukov et al., 2017) that is widely employed for inferring average treatment effects. Since DML does not consider early stopping, we compare it in our toy example only where the decision is made only once. As shown in Section 4.1, such a test cannot detect carryover effects and do not have any power under Example 2.
>
> Third, to implement different reinforcement learning methods, in Appendix B (page 13), we compare our test with a test that integrates the double reinforcement learning (Kallus and Uehara, 2019) and the proposed bootstrap method. We remark that such a test cannot handle the alternating-time-interval and adaptive design, so we compare it in the Markov design only. Meanwhile, other off-policy evaluation methods could be coupled with our bootstrap procedure for A/B testing as well. It can be seen from Figure 3 (page 13) that the resulting test has inflated type-I errors and is less powerful than our proposal.
>
> **Q1: Is $\pi$ deterministic.** Yes, $\pi$ is a deterministic policy. We refer to $\pi$ as the target policy. In the discussion below Lemma 1, the treatments are generated under a behavior policy $b$, potentially different from $\pi$. Our procedure is valid regardless of $b$ (instead of $\pi$) is a deterministic policy or not. We have clarified this in the revision (page 4, line 31-37).
>
> As for the “certain conditions”, please refer to Appendix E (page 15), Conditions C1-C3 as an example. Under these conditions, we show our procedure applies to three difference designs D1-D3 (see Section 4.4, page 6), among which D2 is a deterministic design whereas D1 and D3 are random designs.
>
> **Q2: why the test statistics no longer have the canonical join distribution.** Thanks for this excellent comment! Unlike the settings where observations are independent across time, the test statistics do not have the canonical joint distribution in our setup when adaptive design is used. This is due to the existence of carryover effects in time.
>
> Specifically, when treatment effects are adaptively generated, the behavior policy at difference stages are likely to vary. Due to the carryover effects in time, the state vectors at difference stages have different distribution functions. Consequently, the condition on the covariance matrix of test statistics at different stages is likely to be violated. See Appendix C (page 14) for details.
>
> We have added the related discussions in the main text (page 2, line-15) and relegate the details to Appendix C (page 14).
>
> **Q6: the bandit literature on sequential A/B testing** The paper https://arxiv.org/pdf/1905.11797.pdf is related to the problem of online multiple testing. Specifically, the decision maker observes a sequence of hypotheses arriving in an on-line order. The objective of the decision maker is to decide which of the hypotheses to reject.
>
> The online multiple testing problem is different from our problem of online testing for time-dependent treatment effects. First, there is only one hypothesis in our setup whereas there are multiple hypotheses in online multiple testing. Second, we aim to control the type-I error whereas in online multiple testing, one aims to control the false discovery rate, i.e., the proportion of falsely rejected null hypotheses. Third, the online multiple testing problem usually uses a bandit framework to formulate the problem, whereas we adopt a reinforcement learning framework to characterize the carryover effects in time. Reinforcement learning is different from contextual bandits.
>
> **Q7: inline sketch of proofs.** Thanks for the comment. We have added sketch of proofs of Lemma 1, Theorems 1 and 2 in the paper (page 4 and 6)
>
> **Q8: self-contained definition on some dependent concepts.** Thanks for this question. We have added a self-contained definition of the $\alpha$-spending function approach in the paper (page 2, line-20 to line-18). Specifically, it allocates the total allowable type I error at each interim stage according to an error-spending function.

---

> ### Author Response · Authors · 2020-11-22
> **Response to Reviewer 3 Part 1**
>
> We greatly thank your valuable comments, many of which will lead to a more readable and self-contained version of our paper. We attempt to address all the points in the following. The revised manuscript taking into accounts all your suggestions has been uploaded. Please refer to the most updated revision for details.
>
> In the following, we first address some of your major comments. We next clarify some of your questions.
>
> **Point 1 and Q3, Examples Showcasing Limitations of Existing Tests** Thanks for this excellent comment! Following your suggestion, at the beginning of Section 4 (page 4, line-10), we add the following two toy examples to illustrate the limitations of existing tests:
>
> **Example 1**. $S_t=0.5\varepsilon_t$, $Y_t=S_t+\delta A_t$ for any $t\ge 1$ and $S_0=0.5\varepsilon_0$.
>
> **Example 2**. $S_t=0.5S_{t-1}+\delta A_t+0.5\varepsilon_t$, $Y_t=S_t$ for any $t\ge 1$ and $S_0=0.5\varepsilon_0$.
>
> $\varepsilon_t$'s are independent standard normal random errors. Suppose $\delta>0$. Then $H_1$ holds under both examples. In Example 1, the observations are independent and there are no carryover effects at all. In this case, both the existing A/B tests and the proposed test are able to discriminate $H_1$ from $H_0$. In Example 2, however, treatments have delayed effects on the outcomes. Specifically, $Y_t$ does not depend on $A_t$, but is affected by $A_{t-1}$ through $S_t$. Existing tests will fail to detect $H_1$ as the short-term conditional average treatment effects $\mathbb{E}(Y_t|A_t=1,S_t)-\mathbb{E} (Y_t|A_t=0,S_t)=0$ in this example.
>
> As an illustration, we conduct a small experiment by assuming the decision is made once at $T=500$, and report the empirical rejection probability of the classical two-sample t-test that is commonly used in online experiments, a more complicated nonparametric test based on the double machine learning method (Chernozhukov et al., 2017) that is widely employed for inferring causal effects, and the proposed test. It can be seen the competing methods do not have any power under Example 2 (see Table 1 on page 5, line 1).
>
> We remark that the carryover effects are likely to occur in industries. Take a ridesharing company as an example. As we commented in the introduction, an order dispatching strategy not only affects its immediate income, but also impacts the spatial distribution of drivers in the future, thus affecting its future income. The treatment at a given time can impact both immediate and future outcomes.
>
> Please refer to Section 4.1 (page 4, line-10 to line-1 and page 5, line 1 to line 10) for details.
>
> **Point 2 and Q4, Discussion of more early stopping baselines**
> Thank you very much for this suggestion! We have added these related references in the paper. In addition, the following discussions have been added (page 2, line-12 to line -6):
>
> Recently, there is a growing literature on bringing classical sequential analysis to A/B testing. In particular, Johari et al. (2015) proposed an always valid test based on the classical mixture sequential probability ratio tests (mSPRT). Kharitonov et al. (2015) proposed modified versions of the O'Brien & Fleming and MaxSPRT sequential tests.  Deng et al. (2016) studied A/B testing under Bayesian framework. Abhishek and Mannor (2017) developed a boostrap mSPRT. These tests cannot detect the carryover effects in time, leading to low statistical power in our setup. See the toy examples in Section 4.1 for detailed illustration.

---

### Decision · Program_Chairs · 2021-01-07
**Final Decision**

**Decision:**

Reject

**Comment:**

This paper proposes a testing procedure to determine whether a policy is better than another policy with respect to long-term treatment effects. The reviewers found the problem interesting and saw a lot of value in this work. One of the key concerns was the lack of clarity throughout the paper. The reviews helped the authors actively revise the paper, improving the paper's overall readability throughout the discussion phase. However, the reviewers did not change their ratings. While I agree that this work has merits, since there are many legibility issues, I cannot recommend its acceptance at this stage.